# VAMP2 regulates phase separation of α-synuclein

Aishwarya Agarwal[1,8], Aswathy Chandran[1,8], Farheen Raza[1,6], Irina-Maria Ungureanu[1,7], Christine Hilcenko[1,2,3], Katherine Stott [4], Nicholas A. Bright [1], Nobuhiro Morone [5], Alan J. Warren [1,2,3] & Janin Lautenschläger [1] ✉

α-Synuclein (αSYN), a pivotal synaptic protein implicated in synucleinopathies such as Parkinson's disease and Lewy body dementia, undergoes protein phase separation. We reveal that vesicle-associated membrane protein 2 (VAMP2) orchestrates αSYN phase separation both in vitro and in cells. Electrostatic interactions, specifically mediated by VAMP2 via its juxtamembrane domain and the αSYN C-terminal region, drive phase separation. Condensate formation is specific for R-SNARE VAMP2 and dependent on αSYN lipid membrane binding. Our results delineate a regulatory mechanism for αSYN phase separation in cells. Furthermore, we show that αSYN condensates sequester vesicles and attract complexin-1 and -2, thus supporting a role in synaptic physiology and pathophysiology.

Biomolecular condensation, also known as phase separation, describes the demixing of biomolecules into a highly concentrated dense phase and a depleted dilute phase. The highly condensed phase with weak multivalent interactions offers tight regulation, while the absence of any delimiting membrane facilitates the dynamic exchange of components with the environment[1–4]. By now, biomolecular condensates have been implicated in various complex biological processes ranging from signal transduction to microtubule assembly to gene regulation[5,6]. A series of recent findings have also indicated a role of protein phase separation in synaptic transmission and synaptic vesicle trafficking. For instance, when mixed at an equimolar ratio, the scaffold proteins of the postsynaptic density, PSD-95, GKAP, Shank and Homer undergo phase separation at concentrations well below their synaptic concentrations[7,8]. On-demand release of synaptic vesicles from vesicle clusters can be explained by the fluid-like organization via phase separation of synapsin[9,10]. RIM and RIM-BP, which are components of the presynaptic active zone, have been shown to undergo phase separation in vitro and to form condensates that can effectively cluster voltage-gated calcium channels[11,12]. Furthermore, phase separation of active zone scaffold proteins liprin-α and ELKS-1 is important for recruiting downstream binding partners[13,14]. Finally, phase separation of Eps15 and Fcho1/2, and dynamin, syndapin1 and endophilin has been shown to be involved in endocytosis[15–17].

The presynaptic protein αSYN, involved in neurodegeneration and linked to synucleinopathies such as Parkinson's disease and Lewy body dementia[18] has been reported to undergo protein phase separation[19,20]. Further studies have confirmed these findings demonstrating the formation of early nanoclusters[21] and hardening of αSYN condensates[22,23]. High αSYN concentrations are reached within αSYN droplets, estimated at around 30–40 mM[23]. It has been shown that salt and ions affect αSYN phase separation[24–26] and that αSYN localizes to synapsin condensates in cells[27]. Furthermore, αSYN has been described to co-condensate with tau[28,29]; however, the physiological relevance of αSYN phase separation has not been demonstrated and to date, a clear mechanism on how αSYN phase separation might be regulated is missing.

[1]Cambridge Institute for Medical Research, University of Cambridge, Cambridge Biomedical Campus, Cambridge, UK. [2]Wellcome Trust-Medical Research Council Stem Cell Institute, Jeffrey Cheah Biomedical Centre, Cambridge Biomedical Campus, Cambridge, UK. [3]Department of Haematology, University of Cambridge, School of Clinical Medicine, Jeffrey Cheah Biomedical Centre, Cambridge Biomedical Campus, Cambridge, UK. [4]Department of Biochemistry, University of Cambridge, Cambridge, UK. [5]MRC Toxicology Unit, University of Cambridge, Cambridge, UK. [6]Present address: Protein and Cellular Sciences, GSK, Stevenage, UK. [7]Present address: Department of Clinical Neurosciences, UK Dementia Research Institute, University of Cambridge, Cambridge Biomedical Campus, Cambridge, UK. [8]These authors contributed equally: Aishwarya Agarwal, Aswathy Chandran. ✉e-mail: janin.lautenschlaeger@gmail.com

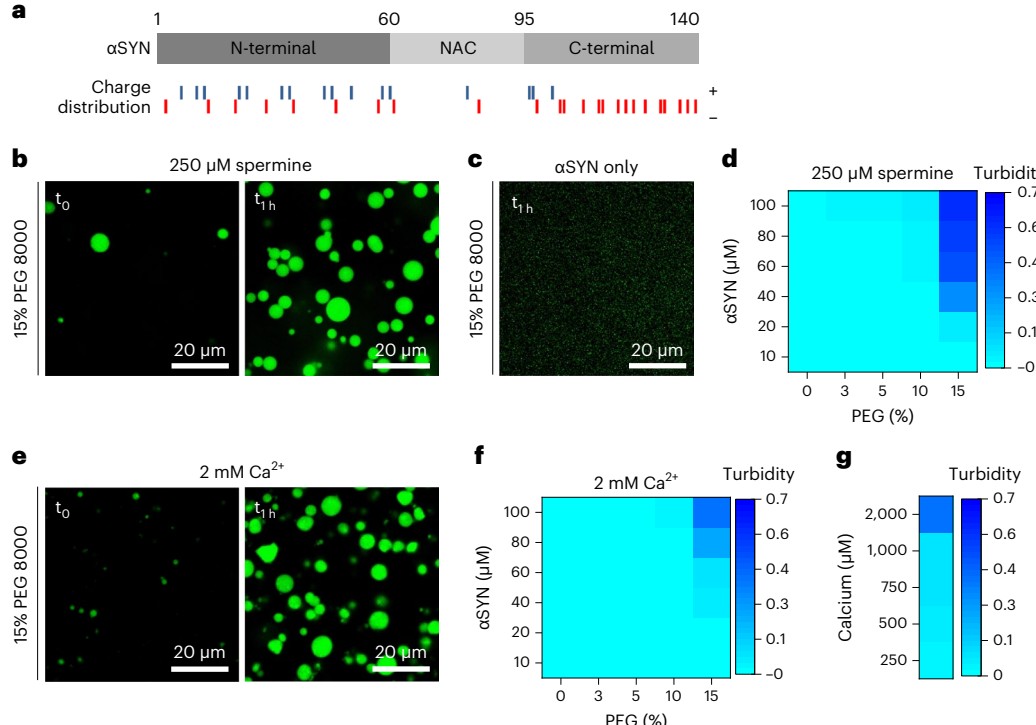

**Fig. 1 | αSYN undergoes phase separation upon electrostatic interaction.**
**a**, Schematic of αSYN showing its three main protein regions, the N-terminal lipid binding region, the NAC region and the negatively charged C terminus. Charge distribution along αSYN sequence; blue indicates positively charged residues and red indicates negatively charged residues. **b**, αSYN phase separation in the presence of 250 μM spermine and crowding with 15% PEG 8000, immediately after PEG addition ($t_0$) and after 1 h ($t_{1h}$). αSYN was used at 100 μM. **c**, αSYN on its own does not show droplet formation in the presence of 15% PEG 8000. αSYN was used at 100 μM. **d**, Heatmap showing turbidity measurements of αSYN phase

separation in the presence of 250 μM spermine. Data represent three biological repeats. **e**, αSYN phase separation in the presence of 2 mM $Ca^{2+}$ and crowding with 15% PEG 8000, immediately after PEG addition ($t_0$) and after 1 h ($t_{1h}$). αSYN was used at 100 μM. **f**, Heatmap showing turbidity measurements of αSYN phase separation in the presence of 2 mM $Ca^{2+}$. Data represent three biological repeats. **g**, Heatmap for αSYN phase separation in the presence of different $Ca^{2+}$ concentrations in the presence of 15% PEG 8000. αSYN was used 100 μM. Data represent three biological repeats.

αSYN constitutes three main regions, the N-terminal domain, the non-amyloid-β (NAC)-region and the negatively charged C terminus (Fig. 1a). The N-terminal domain mediates lipid binding forming an amphipathic α-helix upon interaction with lipid vesicles[30–38]. N-terminal residues 6–25 anchor αSYN to the membrane, while residues 26–97 modulate the strength of its lipid interaction[39,40]. The hydrophobic NAC region has been implicated in self-association and protein aggregation[41–43], whereas the C-terminal region is intrinsically disordered[19] and is neither involved in helix formation[32–37] nor in the formation of αSYN fibrils[44–47]; however, the C-terminal region of αSYN has been shown to interact with synaptic vesicles in the presence of calcium and to influence αSYN localization in synaptosomes[48]. αSYN can bind to multiple protein partners[49] and has been found to interact with the vesicle fusion machinery, in particular the vesicular R-SNARE protein VAMP2 (also known as synaptobrevin-2)[50,51]. In this context, αSYN has been shown to regulate SNARE complex assembly[50,52,53] and synaptic vesicle clustering[54–56].

In this paper, we demonstrate that VAMP2 is involved in the regulation of αSYN phase separation. We first identified that electrostatic interactions at the αSYN C-terminal region modulate αSYN phase separation. We then performed a screen for potential interaction partners and found that VAMP2, which interacts with the C terminus of αSYN[50,51], induces αSYN condensate formation in cells. VAMP2, but not the Q-SNARE proteins syntaxin-1A or SNAP25, induce αSYN condensate formation. Using short peptides of VAMP2, we further show that the juxtamembrane (JM) domain of VAMP2 promotes the phase separation of αSYN. Finally, we show that αSYN condensate formation is dependent on the capacity of αSYN to bind to lipid membranes and

that αSYN condensates accumulate vesicles and complexin-1 and -2 as co-condensation partners. Our results support the role of αSYN phase separation during vesicle cycling, regulated by the R-SNARE VAMP2.

## Results

### αSYN phase separates upon electrostatic interaction

αSYN has been studied for protein aggregation, where divalent cations, such as $Cu^{2+}$ and $Ca^{2+}$ [48,57–59], but also the interference with long-range interactions between its N, NAC and C terminus, have been reported to enhance aggregation[60,61]. To assess whether long-range interactions and electrostatic interactions have a role in αSYN phase separation we tested the potential contribution of spermine and $Ca^{2+}$ on αSYN phase separation. Spermine, a polyamine with four positive charges, which has previously been shown to bind to the negatively charged αSYN C terminus[62] and to break αSYN long-range interactions[60], enabled αSYN to undergo protein phase separation. When αSYN, in the presence of 250 μM spermine, was subjected to crowding mimicked by 15% PEG 8000 rapid droplet formation occurred, with droplet growth over time (Fig. 1b). Under the same conditions, without spermine, αSYN did not show droplet formation (Fig. 1c). To assess αSYN phase separation quantitatively we performed turbidity measurements in the presence of spermine, indicating the crowding conditions and αSYN concentrations at which αSYN undergoes droplet formation (Fig. 1d). Next, we performed droplet and turbidity assays in the presence of $Ca^{2+}$. Similarly, $Ca^{2+}$ did enable αSYN to undergo immediate droplet formation when subjected to crowding with 15% PEG 8000, demonstrating droplet growth over time (Fig. 1e). This was seen in the presence of high $Ca^{2+}$ concentrations (2 mM $Ca^{2+}$); however, in contrast

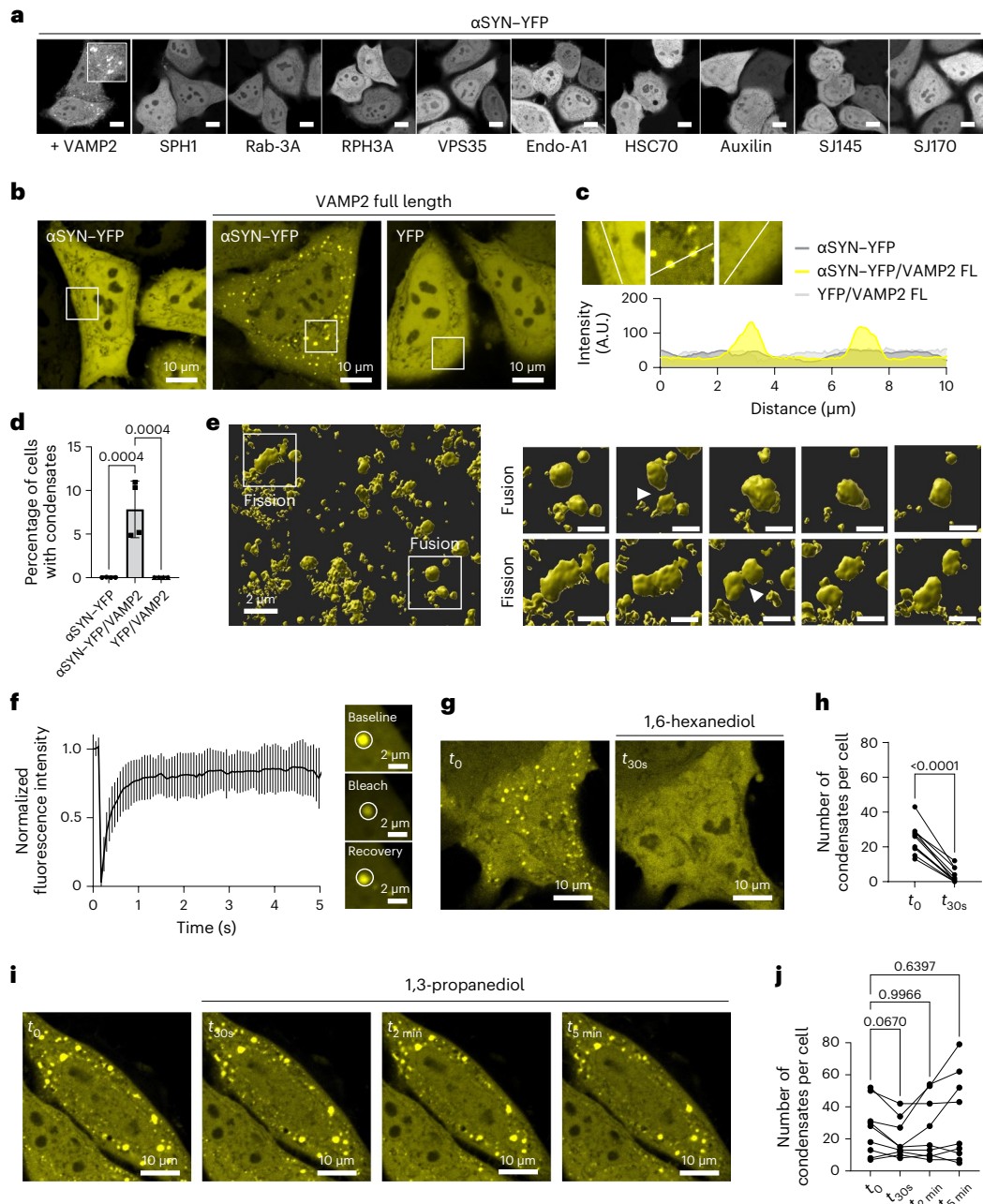

**Fig. 2 | VAMP2 enables αSYN condensate formation in cells. a**, Screening of disease-relevant synaptic proteins on αSYN–YFP distribution upon co-expression in HeLa cells. Scale bar, 20 µm. SPH1, synphilin-1; Rab-3A, Ras-related protein Rab-3A; RPH3A, rabphilin-3A; VPS35, vacuolar protein sorting-associated protein 35; Endo-A1, endophilin-A1; HSC70, heat shock cognate 71 kDa protein; auxilin, putative tyrosine-protein phosphatase auxilin; SJ145 and SJ170, synaptojanin-1 isoforms 1-145 and 1-170. **b**, Cytosolic–nuclear distribution of αSYN–YFP upon ectopic expression in HeLa cells, condensate formation upon co-expression of αSYN–YFP and VAMP2, co-expression of YFP and VAMP2 shows no condensate formation. **c**, Zoomed-in regions and fluorescence intensity distribution for cells with αSYN–YFP only, αSYN with VAMP2 and YFP with VAMP2. FL, full-length. **d**, Quantification of cells forming condensates. Data are derived from Incucyte screening, with 16 images per well, three wells per biological repeat and four biological repeats. *n* indicates biological repeats. Data are mean ± s.d. One-way ANOVA with Dunnett's multiple comparison test. **e**, αSYN–YFP condensates show fluid-like behaviour. Zoomed-in region showing individual fusion event and fission event. Scale bar, 1 µm. See also Supplementary Videos 1–4 and Extended Data Fig. 2. **f**, Photobleaching and recovery of αSYN condensate in cells. Quantification of FRAP experiments. Data are mean ± s.d. with four biological repeats, *n* = 22, n represents individual FRAP experiments. **g**, αSYN–YFP condensates show dispersal upon incubation with 3% 1,6-hexanediol. See also Extended Data Fig. 3 for recovery of αSYN condensates after 1,6-hexanediol washout. **h**, Quantification of staining in **g** before and after incubation with 3% 1,6-hexanediol. *n* = 11 cells, pooled from three biological repeats. The same cells were followed over 30 s. Paired two-tailed *t*-test. **i**, αSYN–YFP condensates are still present after incubation with 3% 1,3-propanediol. **j**, Quantification of staining in **i** before and after incubation with 3% 1,3-propanediol. *n* = 9 cells, pooled from three biological repeats. The same cells were followed over 30 s, 2 min and 5 min. Repeated measures one-way ANOVA with Dunnett's multiple comparison test.

with spermine, no turbidity increase was observed with 10% PEG 8000 (Fig. 1f) or at low $Ca^{2+}$ concentrations in the micromolar range (Fig. 1g). As no droplet formation can be observed at physiologically relevant $Ca^{2+}$ concentrations, which are estimated to be around 200–300 µM $Ca^{2+}$ during synaptic stimulation[63,64], we conclude that the effect of $Ca^{2+}$ is mainly electrostatic. Furthermore, we see αSYN phase separation

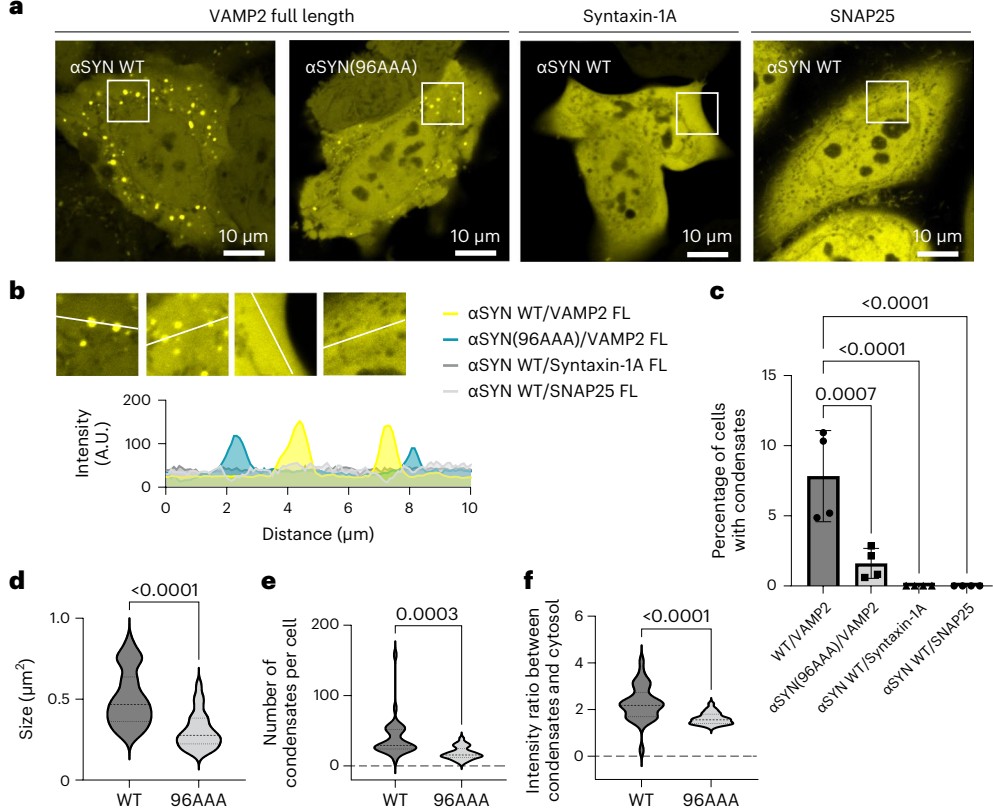

**Fig. 3 | αSYN–VAMP2 interaction regulates αSYN condensate formation.**
**a**, Condensate formation for wild-type (WT) αSYN–YFP and αSYN(96AAA)–YFP upon co-expression with VAMP2. Wild-type αSYN–YFP co-expressed with syntaxin-1A or SNAP25 does not show condensate formation. **b**, Zoomed-in regions and fluorescence intensity distribution for cells with co-expression of wild-type αSYN–YFP with VAMP2 (yellow), αSYN(96AAA)–YFP with VAMP2 (turquoise), and wild-type αSYN–YFP with syntaxin-1A (dark grey) and SNAP25 (light grey). **c**, Quantification of cells forming condensates. Data are derived from Incucyte screening, with 16 images per well, three wells per biological repeat and four biological repeats; $n$ indicates biological repeats. Data are mean ± s.d. One-way ANOVA with Dunnett's multiple comparison test. **d**, Quantification of

condensate size for cells co-expressing wild-type αSYN–YFP and αSYN(96AAA)–YFP with VAMP2. Data are represented as violin plots, $n = 33$ and 26 cells for WT and 96AAA, respectively, pooled from three biological repeats. Unpaired two-tailed $t$-test. **e**, Quantification of condensates per cell for cells co-expressing wild-type αSYN–YFP and αSYN(96AAA)–YFP with VAMP2. mData are represented as violin plots, $n = 33$ and 26 cells for WT and 96AAA, respectively, pooled from three biological repeats. Unpaired two-tailed $t$-test. **f**, Quantification of the intensity ratio between condensates and cytosolic αSYN–YFP for cells co-expressing wild-type αSYN–YFP and αSYN(96AAA)–YFP with VAMP2. Data are represented as violin plots., $n = 33$ and 26 cells for WT and 96AAA, respectively, pooled from three biological repeats. Unpaired two-tailed $t$-test.

only under high crowding conditions, which indicates that further regulatory factors are likely to be involved.

## VAMP2 enables αSYN condensate formation in cells

We next hypothesized that facilitation of αSYN phase separation could occur upon binding of a protein interaction partner. Therefore, we ectopically expressed synaptic proteins that have previously been correlated with Parkinson's disease, together with yellow fluorescent protein (YFP)-tagged αSYN in HeLa cells, evaluating potential condensate formation in live cells. One of the tested proteins, VAMP2, induced αSYN condensate formation in cells (Fig. 2a). αSYN–YFP overexpression on its own shows a cytosolic–nuclear distribution and no sign of αSYN aggregation, congruent with the literature[65,66]; however, when αSYN–YFP was co-expressed with VAMP2, demonstrating simultaneous expression in about 97–100% of cells (Extended Data Fig. 1), a subset of cells manifests clusters of highly concentrated αSYN–YFP. These clusters were not observed upon co-expression of YFP with VAMP2 (Fig. 2b–d).

We performed time-course imaging to distinguish the observed clusters from aggregate structures. We show that the clusters have fluid-like behaviour, demonstrating fusion of separate condensates and condensate fission (Fig. 2e, Supplementary Videos 1–4 and Extended Data Fig. 2). Furthermore, to show mobility between the condensate and the cytosolic αSYN fraction we performed fluorescence recovery

after photobleaching (FRAP) experiments (Fig. 2f). αSYN in condensates showed about 78% recovery 1 s after photobleaching and about 84% recovery 5 s after photobleaching, being congruent with fast recovery reported previously in synapsin/αSYN co-condensates in cells[27]. When the cells were subjected to 1,6-hexanediol, a small aliphatic alcohol[67–71], the clusters showed rapid dispersal (Fig. 2g,h). Furthermore, the dissolution was reversible after brief washout periods (Extended Data Fig. 3). This together demonstrates that the observed clusters are dynamic structures. 1,3-propanediol, a shorter and more hydrophilic aliphatic alcohol, did not show the same effect on αSYN condensate disassembly[72] (Fig. 2i,j).

## αSYN–VAMP2 interaction regulates αSYN condensate formation

The vesicular R-SNARE protein VAMP2 is involved in SNARE complex assembly at the synapse forming a four helical *trans*-SNARE complex with syntaxin-1A and SNAP25, which upon vesicle fusion transitions into the *cis*-SNARE complex, following which, VAMP2 is recycled into vesicles[73–77]. However, VAMP2 has also been shown to be an αSYN interaction partner[50,52,53], where VAMP2 binding occurs at the αSYN C-terminal region[50]. Using alanine scanning of αSYN, the VAMP2-binding site has been mapped to C-terminal residues close to the NAC region[51]. The αSYN(96AAA) mutant, in which αSYN residues 96, 97 and 98 are

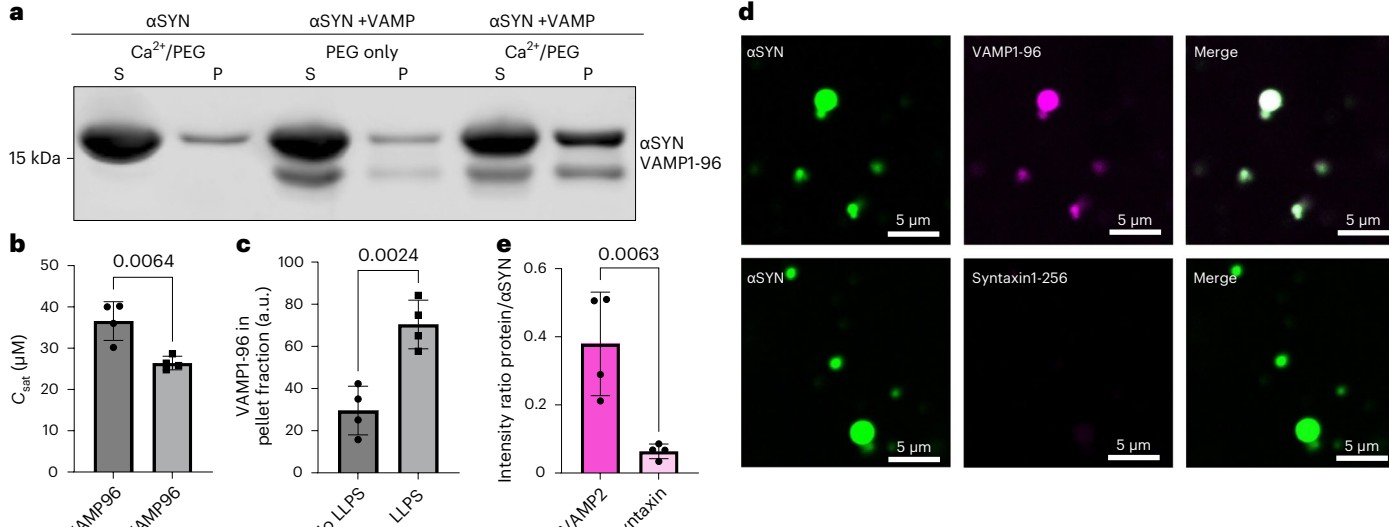

**Fig. 4 | VAMP1-96 promotes αSYN phase separation in vitro. a**, Sedimentation-based assay showing supernatant (S, dilute phase) and pellet (P, droplet phase) fraction upon αSYN phase separation in the presence of $Ca^{2+}$ (concentrations used: 40 μM αSYN, 2 mM $Ca^{2+}$, 15% PEG 8000), upon αSYN and VAMP1-96 incubation (40 μM αSYN, no $Ca^{2+}$, 10 μM VAMP1-96, 15% PEG 8000) and αSYN phase separation in the presence of VAMP1-96 (40 μM αSYN, 2 mM $Ca^{2+}$, 10 μM VAMP1-96, 15% PEG 8000). **b**, Quantification of saturation concentration ($C_{sat}$) of αSYN phase separation in the presence of 2 mM $Ca^{2+}$ (−VAMP) and 2 mM $Ca^{2+}$ with 10 μM VAMP1-96 (+VAMP). Data derived from four biological repeats; $n$ indicates biological repeats. Data are mean ± s.d. Unpaired two-tailed $t$-test. **c**, Quantification for the intensity of VAMP1-96 in the pellet fraction, either under

no phase separation (40 μM αSYN, no $Ca^{2+}$, 10 μM VAMP1-96, 15% PEG) or under αSYN phase separation conditions in the presence of $Ca^{2+}$ (40 μM αSYN, 2 mM $Ca^{2+}$, 10 μM VAMP1-96, 15% PEG). Data derived from four biological repeats; $n$ indicates biological repeats. Data are mean ± s.d. Unpaired two-tailed $t$-test. **d**, Co-localization of VAMP1-96 or syntaxin1-265 with αSYN droplets induced in the presence of 2 mM $Ca^2$ and 15% PEG 8000. Co-localization was evaluated 30 min after induction of αSYN phase separation and addition of the respective labelled protein. αSYN was used at 100 μM. **e**, Quantification of co-localization showing intensity ratio of VAMP1-96 and syntaxin1-265 to αSYN after 30 min. Data derived from four biological repeats; $n$ indicates biological repeats. Data are mean ± s.d. Unpaired two-tailed $t$-test.

replaced with alanine, showed the highest reduction of VAMP2 binding. Therefore, we ectopically co-expressed αSYN(96AAA) with VAMP2 in HeLa cells. The αSYN(96AAA) mutant was able to form condensates (Fig. 3a,b); however, the number of cells forming condensates was significantly reduced (Fig. 3c). Furthermore, the size of condensates and the number of condensates per cell decreased (Fig. 3d,e). In addition, also the intensity ratio between condensate structures and cytosolic αSYN–YFP was reduced, reflecting that less αSYN is taken up into condensates (Fig. 3f). To further test that αSYN condensate formation is specific to its interaction with VAMP2, we also co-expressed αSYN–YFP with the respective synaptic Q-SNAREs. Both Q-SNAREs, syntaxin-1A and SNAP25, did not promote αSYN condensate formation when ectopically expressed with αSYN–YFP in HeLa cells (Fig. 3a–c).

**VAMP1-96 promotes αSYN phase separation in vitro**

Next, we evaluated whether VAMP2 affects αSYN phase separation in vitro. For our assays, we used VAMP1-96, which, without its transmembrane domain, is a soluble protein. To probe whether VAMP1-96 influences αSYN phase separation we estimated the saturation concentration ($C_{sat}$), the concentration at which αSYN phase separation starts, using a sedimentation-based assay. After induction of αSYN droplet formation and centrifugation, the $C_{sat}$ is estimated from the protein level within the supernatant. In the presence of 2 mM $Ca^{2+}$ and 15% PEG 8000 we found a $C_{sat}$ for αSYN phase separation of 33.37 ± 2.36 μM (Fig. 4a,b). The addition of VAMP1-96 decreased the $C_{sat}$ of αSYN phase separation to 26.41 ± 1.64 μM (Fig. 4a,b). In addition, we observed that VAMP1-96 is pulled down into the pellet fraction when αSYN phase separation is induced, indicative of its recruitment to αSYN droplets (Fig. 4a (lane 4/6) and 4c). Therefore, we next probed whether VAMP1-96 would co-condense with αSYN droplets. αSYN phase separation was induced in the presence of 2 mM $Ca^{2+}$ and 15% PEG 8000 as in our previous experiments. The reaction was supplemented with labelled

VAMP1-96, demonstrating co-localization of VAMP1-96 with αSYN droplets (Fig. 4d,e). As a control, we used the Q-SNARE syntaxin-1A, again as a soluble protein without its transmembrane domain. Labelled syntaxin1-265 showed significantly less uptake into αSYN droplets (Fig. 4d,e), demonstrating that co-localization is specific for VAMP1-96.

**The JM domain of VAMP2 enables αSYN phase separation**

We next tested which region of VAMP2 is involved in inducing αSYN phase separation. We focused on two regions that have been previously described to interact with αSYN, the N-terminal domain of VAMP2 (ref. 50), but also a more C-terminal region that has been shown to mediate αSYN–VAMP2 binding during aggregation[78]. For this we used peptides, one identical to residues 25–30 of the VAMP2 N-terminal domain and one identical to residues 83–88 in the JM domain of VAMP2 (Fig. 5a). In the droplet as well as in the turbidity assay the N-terminal peptide (NT peptide, NLTSNR) did not influence αSYN phase separation (Fig. 5b,c); however, the JMD peptide (KLKRKY) was found to induce αSYN droplet formation (Fig. 5b) and mediated increased turbidity (Fig. 5c). To test specificity and to cover larger protein regions, we designed longer peptides, identical to either the first half of the N-terminal domain (NT long peptide 1, amino acids (aa) 3–16, ATAATAPPAAPAGE), the second half of the N-terminal domain (NT long peptide 2, aa 17–30, GGPPAPPPNLTSNR), the full-length JM domain (JMD long peptide, aa 83–96, KLKRKYWWKNLKMM) or the SNARE–JMD interlinking region including the residues of the previous short JMD peptide (SNARE–JMD long peptide, aa 75–88, SQFETSAAKLKRKY; Fig. 5d). The droplet assay and the turbidity measurements show that the NT peptides and the SNARE–JMD long peptide do not induce αSYN phase separation, while the JMD long peptide enabled αSYN droplet formation indicating the role of the JM domain in promoting αSYN condensate formation (Fig. 5e,f). Furthermore, to test the sequence-specific effect on phase separation, we designed a scrambled JMD long peptide (Peptidenexus

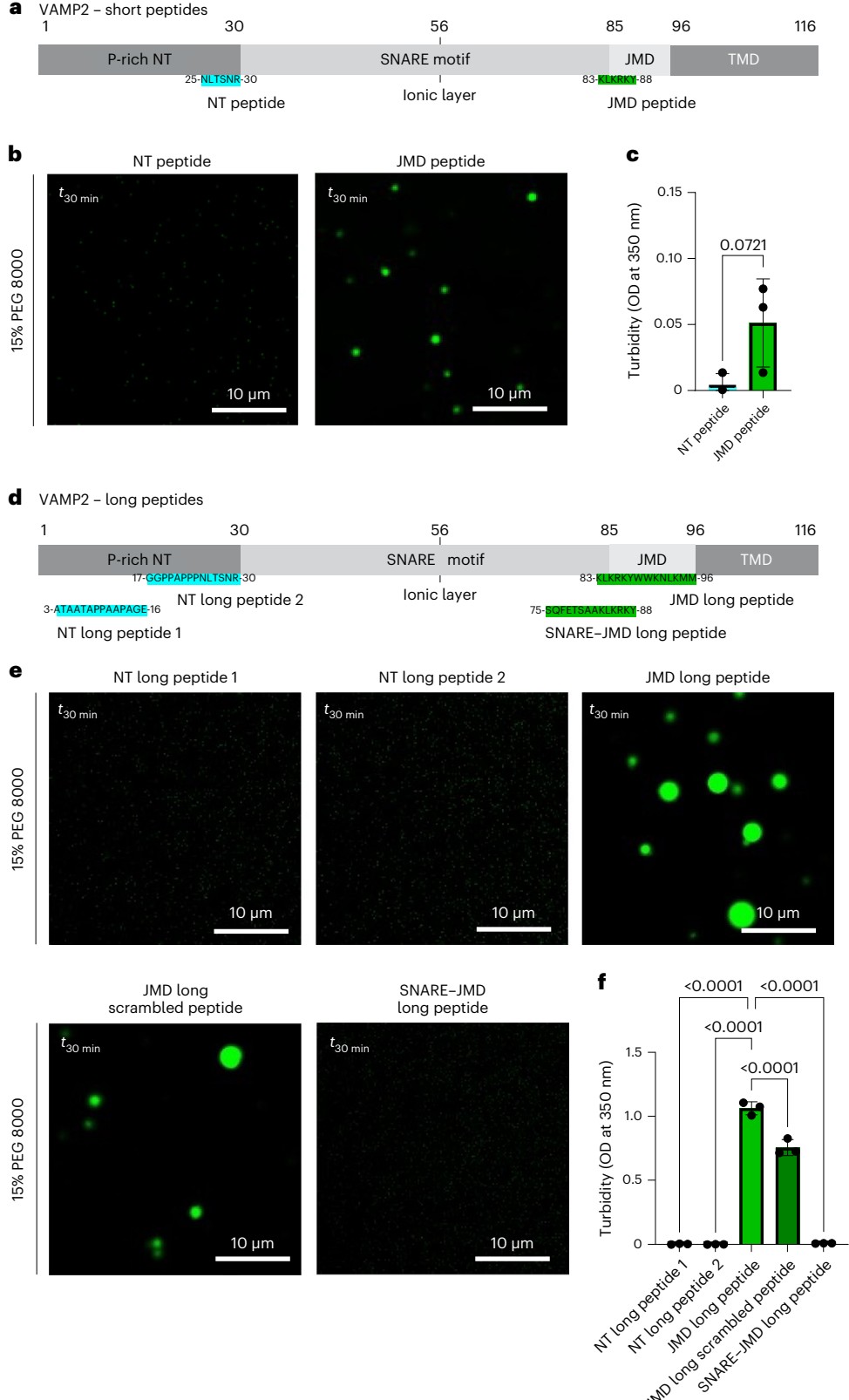

**Fig. 5 | The JM domain of VAMP2 enables αSYN phase separation. a**, Schematic of VAMP2 with the respective NT peptide and JMD peptide. **b**, Evaluation of αSYN phase separation in the presence of 150 μM peptide and crowding with 15% PEG 8000, 30 min after PEG addition ($t_{30\,min}$), 40 μM αSYN. **c**, Quantification of turbidity measurements at $t_{30\,min}$. Data are mean ± s.d. from three biological repeats; $n$ indicates biological repeats. Unpaired two-tailed $t$-test. **d**, Schematic of VAMP2 with the respective N-terminal long peptides (NT long peptide 1 and 2), the native JMD long peptide and the SNARE–JMD long peptide. **e**, Evaluation of αSYN phase separation in the presence of 150 μM peptide and crowding with 15% PEG 8000, 30 min after PEG addition ($t_{30\,min}$), 40 μM αSYN. **f**, Quantification of turbidity measurements at $t_{30\,min}$. Data are mean ± s.d. from three biological repeats; $n$ indicates biological repeats. One-way ANOVA with Dunnett's multiple comparison test.

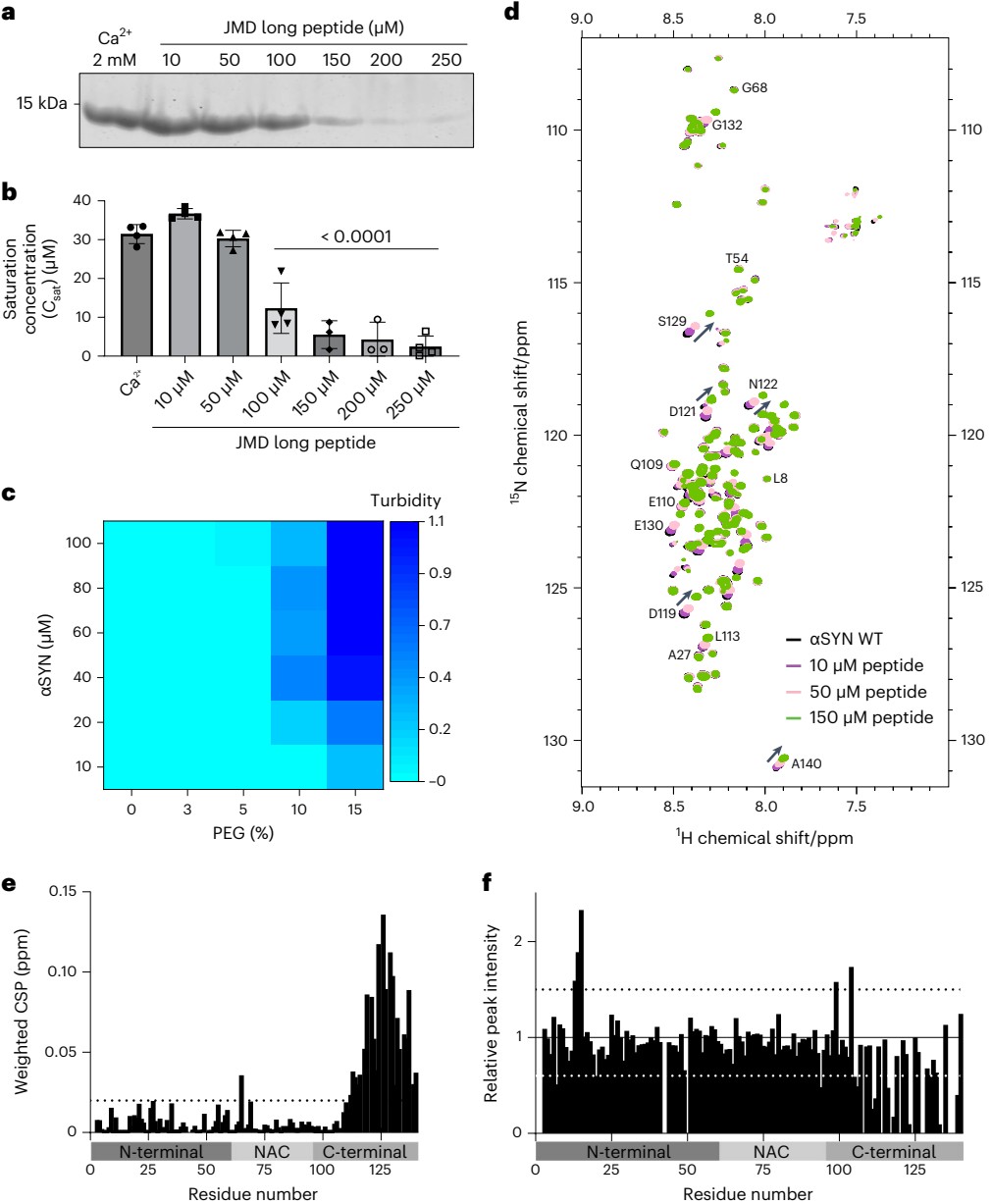

**Fig. 6 | The JM domain of VAMP2 interacts with αSYN. a**, Sedimentation-based assay showing supernatant (dilute phase) upon αSYN phase separation in the presence of the JMD long peptide (40 μM αSYN, 15% PEG 8000, concentrations of peptide as indicated). **b**, Quantification of $C_{sat}$ of αSYN phase separation in the presence of the JMD long peptide. Data are mean ± s.d., $n$ represents three biological repeats for 150 μM and 200 μM peptide and four biological repeats for all other conditions. One-way ANOVA with Dunnett's multiple comparison test. **c**, Heatmap showing turbidity measurements of αSYN phase separation in the presence of 150 μM JMD long peptide. Data represent three biological repeats. See also Extended Data Fig. 4 for αSYN droplet formation in the presence of 150 μM JMD long peptide at lower PEG concentrations. **d**, Overlapped $^1$H-$^{15}$N-BEST-TROSY spectra of αSYN without (black) and in the presence of increasing

concentrations of JMD long peptide as indicated. Also see Extended Data Fig. 5a for the $^1$H-$^{15}$N-BEST-TROSY spectra of αSYN in the presence of NT long peptide 1. αSYN concentration was 40 μM and peptide concentrations were 10 μM, 50 μM and 150 μM. **e**, Weighted average (of $^{15}$N and $^1$H) chemical shift perturbation ($\Delta\delta = \sqrt{(\delta H^2 + 0.2\,(\delta^{15}N)^2)}$) of residues in αSYN in the presence of 150 μM JMD long peptide. Also see Extended Data Fig. 5b weighted average (of $^{15}$N and $^1$H) chemical shift perturbation in the presence of NT long peptide 1. Dashed line indicates weighted CSP of 0.02 ppm. **f**, Relative peak intensities of αSYN residues in the presence of 150 μM JMD long peptide. Also see Extended Data Fig. 5c for relative peak intensities of αSYN residues in the presence of NT long peptide 1. The black dashed line indicates a relative peak intensity of 1.5 and the white dashed line indicates a relative peak intensity of 0.6.

scrambler, KMLKWKMNKYLRWK). The native JMD peptide showed a significantly stronger effect than the scrambled JMD long peptide, demonstrating that the effect on αSYN phase separation is not solely mediated by the number of charges.

## The JM domain of VAMP2 interacts with αSYN

To further validate the effect of the JMD long peptide we estimated the $C_{sat}$ for αSYN phase separation in the presence of peptide using

the sedimentation-based assay as described above. A significant decrease in αSYN $C_{sat}$ was seen from 100 μM JMD long peptide, with a robust reduction starting from 150 μM JMD long peptide. Here $C_{sat}$ levels were between 2 and 9 μM (Fig. 6a,b). Using 150 μM JMD long peptide, we next performed turbidity measurements, which demonstrate that αSYN phase separation is induced already at 10 μM αSYN in the presence of 15% PEG 8000 and at 20 μM αSYN in the presence of 10% PEG 8000 (Fig. 6c). Droplet formation in the presence of JMD

long peptide was observed at 10% and also 5% PEG 8000 (Extended Data Fig. 4).

To test whether and where the JMD long peptide interacts with αSYN we applied multi-dimensional NMR spectroscopy[79,80], performing chemical shift perturbation titrations on αSYN using the JMD long peptide at concentrations estimated above (Fig. 6d). The NMR data demonstrate that αSYN is intrinsically disordered as shown previously[32,48,60,62,81]. Upon addition of the JMD long peptide, we observed chemical shift perturbations and decreased relative peak intensities that were most prominent in the αSYN C-terminal region (Fig. 6e,f), indicating that the JMD long peptide interacts with the negatively charged C-terminal domain or that the environment of these residues changes due to a conformational shift. We also observed smaller chemical shift perturbations and altered relative peak intensities at residues throughout the αSYN N-terminal and NAC regions, with increased peak intensities at residues E13, G14, V15 and Q99, and E104 (relative peak intensity >1.5) and decreased peak intensities at residues G7, E20 and F94 (relative peak intensity <0.6). These additional changes support the idea that αSYN undergoes a conformational change upon peptide binding, aligning with previous reports indicating long-range intramolecular interactions within αSYN[60,62,82,83], and with molecular dynamics simulations of αSYN reported recently[84]. The modulation of these interactions could contribute to the enhancement of αSYN condensate formation. Multi-dimensional NMR spectroscopy[79,80] was also performed with the NT long peptide which does not affect αSYN phase separation, but here no chemical shift perturbations or relative peak intensity changes were observed in the respective NMR spectrum (Extended Data Fig. 5a–c).

### αSYN(A30P) shows no condensate formation in cells

As both, VAMP2 via its transmembrane domain and αSYN, via the formation of an amphipathic α-helix, bind to lipid membranes, we next hypothesized that αSYN condensate formation occurs on lipid membranes. To test the relevance of lipid binding we used αSYN(A30P), an αSYN disease variant[85] with defective lipid binding[31,38,86–88]. Upon co-expression of αSYN(A30P)–YFP with VAMP2 no condensate formation was observed (Fig. 7a,b). This finding suggests that the lipid membrane association of αSYN is important to allow αSYN condensate formation. It is noteworthy that αSYN(A30P) has reduced but not completely abolished lipid binding, being diminished to about half of wild-type αSYN binding[87]. Therefore, reduced rather than abolished condensate formation might have been expected. Our results, however, indicate that in cells αSYN(A30P) might not reach the critical concentration at lipid membranes to nucleate αSYN phase separation. This is consistent with previous findings on the membrane accumulation of αSYN(A30P) in yeast cells and neurons[89,90]. Wild-type αSYN and αSYN(A53T) are associated with the plasma membrane when expressed in yeast, whereas αSYN(A30P) shows a diffuse cytoplasmic

distribution[89]. Similarly, αSYN(A30P) behaves like a fully soluble protein upon photobleaching in synaptic terminals, whereas wild-type αSYN recovers more slowly, indicative of its membrane-bound fraction. Furthermore, upon synaptic stimulation, there is no redistribution seen for αSYN(A30P), whereas wild-type αSYN, similar to synapsin, dissociates upon exocytosis[90].

### αSYN condensates assemble vesicles

Further to these experiments we examined whether αSYN condensates have a role in vesicle clustering, such as found upon formation of synapsin condensates in cells[91]. We therefore co-expressed synaptotagmin, a vesicle-bound transmembrane protein involved in Ca[2+] sensing, together with αSYN–YFP and VAMP2. Although mScarlet synaptotagmin showed small vesicular structures on its own, it also demonstrates co-localization with the αSYN condensates (Fig. 7c,d). We found that synaptotagmin within condensates shows lower fluorescence intensity compared with vesicular structures outside condensates (Fig. 7e,f), indicating that αSYN condensates attract these vesicles over time. To further investigate whether αSYN condensates contain vesicles we conducted correlative light and electron microscopy. Combining fluorescence imaging of condensates with ultrastructural visualization through electron microscopy, we demonstrate that αSYN condensates each harbour approximately 10–50 vesicles. Notably, the vesicle clustering seen for αSYN is distinct from vesicle clustering observed in synapsin-1 condensates[91,92], demonstrating a more mixed population of vesicles including large vesicles with diameters of around 500 nm in diameter, smaller vesicles with sizes around 100–150 nm and double-membrane autophagosomes (Fig. 7g–i and Extended Data Fig. 6). Therefore, our findings indicate a complementary function of αSYN condensates and synapsin-1 condensates in mediating vesicle clustering, where both proteins could attract different vesicles or act at different steps during vesicle re-clustering.

### αSYN condensates co-condense complexin-1 and -2

Further to this, condensates are known to convey functional roles by attracting protein partners[6,93,94]. We here reveal that αSYN condensates specifically enrich complexin-1 and complexin-2, synaptic proteins modulating vesicle release[95–97]. We find that both complexin-1 and complexin-2 are enriched compared with mScarlet control; however, complexin-2, predominantly expressed in excitatory neurons, is more concentrated compared with complexin-1 (Fig. 7j,k), mainly expressed in inhibitory neurons[98–100]. This pattern aligns with a predominant expression of αSYN in excitatory neurons[101]. These findings are particularly notable as complexin upregulation has been observed in αSYN knockout and mutant mice[102,103]. Furthermore, complexin has been linked to human disease, such as Parkinson's disease and other synucleinopathies[104,105]. Here, complexin-1 has been identified as a Parkinson's disease risk factor[104], and complexin-2 levels were reported to

---

**Fig. 7 | αSYN condensate formation is dependent on αSYN lipid membrane binding and attracts vesicles and protein partners. a**, Co-expression of VAMP2 and wild-type αSYN–YFP in HeLa cells showing condensate formation. Cells co-expressing VAMP2 and αSYN(A30P)–YFP lack condensate formation. **b**, Quantification of cells forming condensates. Data are derived from Incucyte screening, with 16 images per well, three wells per biological repeat and four biological repeats. *n* indicates biological repeats. Data are mean ± s.d. One-way ANOVA with Dunnett's multiple comparison test. **c**, Co-expression of αSYN–YFP, VAMP2 and mScarlet synaptotagmin showing partial co-localization of mScarlet synaptotagmin with αSYN–YFP condensates. **d**, Quantification of Pearson correlation coefficient for αSYN–YFP and mScarlet synaptotagmin co-localization. Three biological repeats were conducted; *n* indicates biological repeats. Data are mean ± s.d. Unpaired two-tailed *t*-test. **e**, Zoomed-in areas highlighting co-localization of αSYN condensates with co-expressed mScarlet synaptotagmin and mScarlet synaptotagmin outside αSYN condensates. Fluorescence intensity distribution for αSYN–YFP (yellow) and mScarlet

synaptotagmin (magenta). **f**, Quantification of mScarlet synaptotagmin intensity outside and within αSYN condensates. *n* = 10 cells, pooled from four biological repeats. Data are mean ± s.d. Unpaired two-tailed *t*-test. **g**, HeLa cells with ectopic expression of αSYN–YFP, VAMP2 and 4xMTS-mScarlet, electron microscopy image overlaid with fluorescence microscopy, showing assemblies of vesicles colocalizing with αSYN–YFP condensates. Also see Extended Data Fig. 6 for individual images. **h**, Electron microscopy images for individual vesicle clusters in Fig. 7g. Scale bar, 1 μm. **i**, Histogram showing size distribution of vesicles contained within αSYN condensates. Data are mean ± s.d. *n* = 14 vesicle clusters pooled from two cells from two biological repeats. **j**, Co-expression of αSYN–YFP, VAMP2 and complexin-1/2 mScarlet demonstrating enrichment of complexins within αSYN–YFP condensates. **k**, Quantification of complexin in αSYN condensates versus cytosolic complexin levels. Complexin-2 levels were significantly higher than complexin-1. *n* = 21 and 23 cells for complexin-1 and complexin-2, respectively, pooled from three biological repeats. Data are mean ± s.d. Unpaired two-tailed *t*-test.

be decreased in multiple system atrophy[105]. In summary, these findings highlight a potential functional interplay with complexin that warrants further investigation in the context of αSYN phase separation.

## Discussion

With the concept of protein phase separation new perspectives for our understanding of cell compartmentalization and on how intracellular processes are organized in space and time emerge[1,5,106]. This is true for cytosolic and nuclear phase-separated compartments, but also for the organization within the presynaptic terminal[9–17,107].

For the presynaptic protein αSYN the concept of biomolecular condensates opens new avenues for understanding its role at the synapse, as, besides its clear link to disease[18,85,108], its normal physiological function remains elusive. To date, several reports show that αSYN can undergo droplet formation[19,20,24,25,109] and that these condensates show hardening, which is potentially relevant for the transition to pathological states[22,23]. Here, we demonstrate that the phase separation of αSYN can be regulated via C-terminal interactions. We find that the presence of spermine and also Ca$^{2+}$ promotes fast droplet formation of αSYN under otherwise identical conditions. This aligns with recent studies

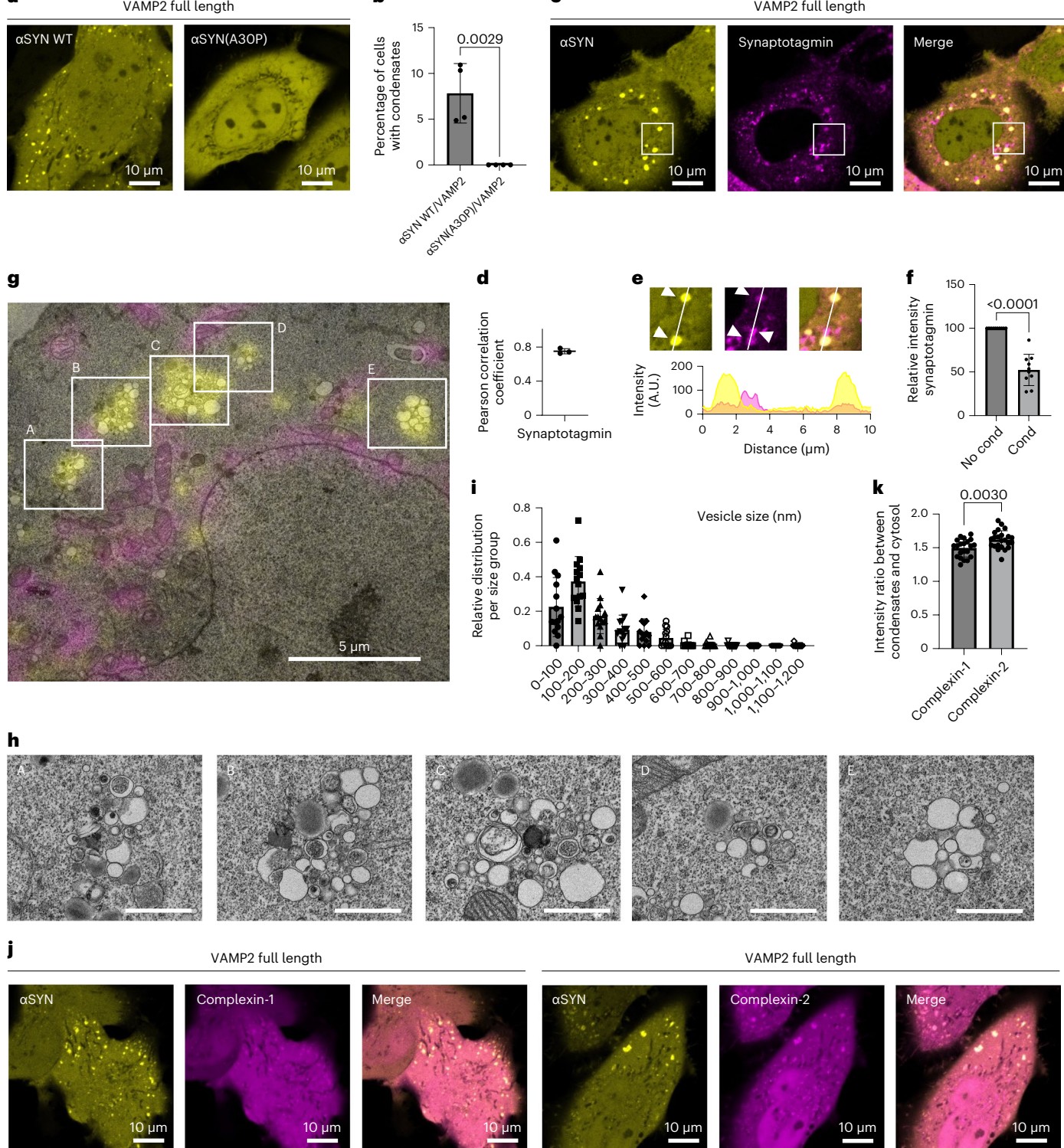

demonstrating a facilitatory effect of Ca[2+] and Mn[2+] on αSYN phase separation[25,26]; however, we do not observe facilitation of αSYN droplet formation at physiologically relevant Ca[2+] concentrations estimated to be 200–300 µM Ca[2+] or even below during synaptic stimulation[63,64]. Therefore we speculate that this reflects an electrostatic modulation, rather than a direct regulatory role of Ca[2+] itself. We demonstrate that in cells, co-expression of αSYN with an interaction partner, namely VAMP2, leads to the formation of αSYN condensates. αSYN has been shown before to interact with VAMP2, in particular via its C-terminal region[50–52]; however, a role for the induction of αSYN phase separation has not been shown before. Our findings on the role of VAMP2 on αSYN phase separation might reflect earlier reports that indicate that αSYN dispersion upon synaptic activity is inhibited when SNARE-mediated synaptic vesicle exocytosis is blocked[90,110]. The experiments conducted emphasize that not Ca[2+] entry but rather vesicle exocytosis itself and potentially a conformational change of the individual SNARE proteins can influence the spatiotemporal organization of αSYN, aligning with our findings here.

In addition to the role of VAMP2 on αSYN phase separation, our results demonstrate that the formation of αSYN condensates in cells is dependent on αSYN binding to lipid membranes. Here, the αSYN(A30P) variant with decreased lipid binding completely abolished condensate formation. This indicates that αSYN forms membrane-associated condensates rather than undergoing droplet formation in solution, which is in accordance with its well-described vesicle binding property[30–38]. Notably, the A30P variant abolishes αSYN condensate formation completely, whereas in vitro binding of αSYN(A30P) is only reduced by about half[87,111]. Again, this aligns with previous findings, where αSYN(A30P) completely abolishes the membrane accumulation of αSYN in yeast and at the synapse[89,90]. Further to that, we show that αSYN condensates in cells cluster vesicles. In contrast to the vesicle clusters that have been found upon synaptophysin/synapsin co-expression[91], we found that vesicles of different sizes are maintained within αSYN condensates, indicating a differential function of αSYN in vesicle homoeostasis. In this context, different vesicle cluster entities formed by distinct condensates have been demonstrated lately[92]. Recent findings in the lamprey giant reticulospinal synapse verify a role of αSYN in synaptic vesicle clustering directly at the synapse[112], supporting our findings and previous in vitro experiments on αSYN vesicle clustering[54–56]. In the lamprey synapse, αSYN depletion not only affected the distal pool of vesicles[112], as has been found upon synapsin depletion or interference with synapsin phase separation[10,113], but also the proximal vesicle pool located adjacent to the active zone, again emphasizing different and/or supplementary roles for αSYN and synapsin[27,112].

Our data show that short peptides identical to the JM domain of VAMP2 are able to initiate αSYN phase separation. Peptides identical to the N-terminal domain of VAMP2, however, did not promote αSYN phase separation in vitro. This finding was surprising as immunoprecipitation experiments show that the N-terminal proline-rich domain of VAMP2 mediates VAMP2–αSYN interaction[50]. Though, in the context of the full-length protein, our data obtained with short peptides, might not reflect the full picture; it has been shown that small proteins containing only one interaction site can act as a cap and prevent condensate formation as no network interactions are promoted[114]. Therefore, with the data we have, we do not exclude a role for the N-terminal proline-rich region but suggest that the positively charged JM domain has an additional role mediated via electrostatic interaction with the negatively charged C-terminal domain of αSYN. Other transmembrane proteins, such as syntaxin-1A, exhibit similar positively charged motifs within their stop transfer signal. We here demonstrate that VAMP2 specifically induced αSYN phase separation, but other VAMP family members might have similar effects on αSYN, which is indeed likely as αSYN overexpression has been shown to interfere with other vesicle transport mechanisms[115,116]. Sterically the JM domain of VAMP2 is in close proximity to the lipid membrane as well as the αSYN residues,

which have been demonstrated to interact most strongly with VAMP2 (ref. 51). An additional regulatory role for the proline-rich N-terminal region of VAMP2 as well as the more C-terminal residues of αSYN, via S129 phosphorylation and attracting or modulating further protein partners, is to be explored[84,117].

In summary, our results demonstrate that αSYN phase separation is regulated via its C-terminal domain. This can be achieved via electrostatic interactions between the negatively charged C terminus and positively charged molecules, such as spermine or Ca[2+], or via protein–protein interactions with VAMP2, in particular its JM domain. While Ca[2+] seems to modulate αSYN phase separation only at high concentrations, the JMD peptide allowed induction of αSYN phase separation at a concentration that aligns with the estimated VAMP2 concentration at the synapse, which is reported to be 170 µM[118]. We demonstrate that αSYN phase separation is linked to VAMP2 and therefore could be influenced upon SNARE complex formation and during the synaptic vesicle cycle. Further to this, we find that αSYN condensates cluster vesicles and attract other protein partners. The vesicle clustering mediated by αSYN is distinct from vesicle clustering observed in synapsin-1 condensates and might indicate a wider role in capturing vesicles. Our findings delineate a molecular mechanism for the regulation of αSYN phase separation, which will allow further exploration of its role during vesicle cycling.

## Online content

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

## Methods

### Protein expression and purification of αSYN

Recombinant wild-type human full-length αSYN was expressed in BL21(DE3) competent *Escherichia coli* (C2527, NEB) using vector pET28a (Addgene 178032). Bacteria were cultured in LB medium supplemented with 50 µg ml⁻¹ kanamycin (37 °C, constant shaking at 250 rpm). Expression was induced at an $OD_{600}$ of 0.8 using 1 mM isopropyl β-ᴅ-1-thiogalactopyranoside (IPTG) and cultured overnight at 25 °C. Cell pellets were collected by centrifugation at 4,000$g$ for 30 min (AVANTI J-26, Beckman Coulter). αSYN was purified using a protocol previously described[119]. In brief, the cell pellet was resuspended in lysis buffer (10 mM Tris, 1 mM EDTA, Roche cOmplete EDTA-free protease inhibitor cocktail, pH 8). The cells were disrupted using a cell disruptor (Constant Systems) and were ultracentrifuged at 4 °C, 186,000$g$ for 20 min (Ti-45 rotor, Optima XPN 90, Beckman Coulter). The supernatant was collected and heated for 20 min at 70 °C to precipitate heat-sensitive proteins, followed by ultracentrifugation as above. Streptomycin sulfate (5711, EMD Millipore) was added at a final concentration of 10 mg ml⁻¹ to the supernatant and continuously stirred at 4 °C for 15 min to precipitate DNA, followed by ultracentrifugation as above. Ammonium sulfate (434380010, Thermo Scientific) was added at a final concentration of 360 mg ml⁻¹ to the supernatant and continuously stirred at 4 °C for 30 min to precipitate the protein. The precipitated protein was then centrifuged at 500$g$ for 15 min, dissolved in 25 mM Tris, pH 7.7 and dialysed overnight against the same buffer to remove salts. The protein was purified using ion exchange on a HiTrap Q HP 5-ml anion exchange column (17115401, Cytiva) using gradient elution with 0–1 M NaCl in 25 mM Tris, pH 7.7. The collected protein fractions were run on SDS–PAGE and pooled fractions were further purified using size-exclusion chromatography on a HiLoad 16/600 Superdex 75-pg column (28989333, Cytiva). The fractions were collected, and their purity was confirmed using SDS–PAGE analysis. Protein concentrations were determined by measuring absorbance at 280 nm using an extinction coefficient of 5,600 M⁻¹cm⁻¹. The monomeric protein was frozen in liquid nitrogen and stored in 25 mM HEPES buffer, pH 7.4, at −70 °C. pET28a Cdk2ap1CAN was a gift from L. He (Addgene plasmid 178032; RRID: Addgene_178032)[120].

### Protein expression and purification of VAMP1-96 and syntaxin1-265

Recombinant wild-type human VAMP1-96 and syntaxin1-265 were expressed in BL21(DE3) competent *Escherichia coli* (C2527, NEB) using vector pOPINS and pET28a (Addgene 66711 and 178032), respectively. Bacteria were cultured in LB medium supplemented with 50 µg ml⁻¹ kanamycin (37 °C, constant shaking at 250 rpm). Expression was induced at an $OD_{600}$ of 0.8 using 1 mM IPTG and cultured overnight at 25 °C. Cell pellets were collected by centrifugation at 4,000$g$ for 30 min (AVANTI J-26, Beckman Coulter). HisSUMO-tagged VAMP1-96 was purified using Ni-NTA chromatography using the following protocol. In brief, the cell pellet was resuspended in lysis buffer (25 mM HEPES, 300 mM NaCl, Roche cOmplete EDTA-free protease inhibitor cocktail, pH 7). The cells were disrupted using a cell disruptor (Constant Systems) and were ultracentrifuged at 4 °C, 186,000$g$ for 20 min (Ti-45 rotor, Optima XPN 90, Beckman Coulter). The supernatant was incubated with Ni-NTA resin overnight at 4 °C, which was then loaded on the column. The column was washed with 25 mM imidazole and the His-tagged protein was eluted with 250 mM imidazole, pH 7. The eluted protein was incubated with SUMO protease (10 U mg⁻¹ protein, SAE0067, Sigma) and was set for overnight dialysis (25 mM HEPES, 300 mM NaCl, pH 7) at 4 °C. SUMO protease and uncleaved protein were removed incubating the dialysed protein solution with Ni-NTA resin for 2 h at 4 °C. The flow-through with His-cleaved VAMP1-96 protein was collected and its purity was confirmed using SDS–PAGE analysis. Protein concentrations were determined by measuring absorbance at 280 nm using an extinction coefficient of 13,980 M⁻¹cm⁻¹. The

monomeric protein was frozen in liquid nitrogen and stored at −70 °C. His-tagged syntaxin1-265 was purified using Ni-NTA chromatography as described above. In brief, the cell pellet was resuspended in lysis buffer (25 mM HEPES, 300 mM NaCl, 1 mM dithiothreitol (DTT), Roche cOmplete EDTA-free protease inhibitor cocktail, pH 7.4). The supernatant obtained after cell disruption and centrifugation was directly loaded on to the Ni-NTA column. The column was washed with 25 mM imidazole and the His-tagged protein was eluted with 250 mM imidazole and was dialysed against buffer (25 mM HEPES, 300 mM NaCl, 1 mM DTT, pH 7.4) overnight. The protein was further purified using size-exclusion chromatography on a HiLoad 16/600 Superdex 200-pg column (28989335, Cytiva). The fractions were collected, and their purity was confirmed using SDS–PAGE analysis. Protein concentrations were determined by measuring absorbance at 280 nm using an extinction coefficient of 7,450 M⁻¹cm⁻¹. The monomeric protein was frozen in liquid nitrogen and stored at −70 °C. pOPINS-UBE3C was a gift from D. Komander (Addgene plasmid 66711; RRID: Addgene_66711)[121]. pET28a Cdk2ap1CAN was a gift from L. He (Addgene plasmid 178032; RRID: Addgene_178032)[120].

### Protein labelling

Labelling of proteins was performed in bicarbonate buffer (C3041, Sigma) at pH 8 using NHS-ester active fluorescent dyes. VAMP1-96 and syntaxin1-256 were labelled using Janelia Fluor 549 SE (6147, Tocris), αSYN was labelled using Alexa Fluor 488 5-SDP ester (A30052, Invitrogen Thermo Fisher). Excess-free dye was removed by buffer exchange using PD10 desalting columns (IP-0107-Z050.0-001, emp BIOTECH, Generon). Labelled protein concentrations were estimated using molar extinction coefficients of the dyes, $\varepsilon_{555\,nm} = 101,000$ M⁻¹cm⁻¹ for Janelia Fluor 549 SE; $\varepsilon_{494\,nm} = 72,000$ M⁻¹cm⁻¹ for Alexa-488 5-SDP ester.

### Phase separation assays including turbidity measurements, confocal imaging and sedimentation-based assays

Phase separation assays were performed in 25 mM HEPES, pH 7.4 unless mentioned otherwise. Phase separation was induced by mixing αSYN and PEG 8000 (BP223, Fisher Bioreagent) in the presence or absence of calcium (21108, Sigma), spermine (S2876, Sigma) or VAMP2 peptides (custom synthesis with Proteogenix, Schiltigheim) as indicated respectively. For turbidity measurements, phase-separated samples were set up as described above. The turbidity of the samples was measured at 350 nm, 25 °C using 96-well Greiner optical bottom plates on a CLARIOstar plate reader (BMG LABTECH) under quiescent conditions. CLARIOStar 5.01 was used for data acquisition. A sample volume of 100 µl was used, and readings were taken within 5 min of sample preparation. For phase diagrams, the raw turbidity data are plotted with background subtraction. Data were obtained from at least three independent sets of biological samples and were plotted using OriginPro 2018. For confocal microscopy, images for phase-separated samples were acquired on an LSM780 confocal microscope (Zeiss) using a 63× oil immersion objective. Zen 2.3 (black edition) and Zen 2.6 (blue edition) were used for data collection. Images were taken in brightfield mode and/or using αSYN supplemented with 1% Alexa 488 labelled αSYN. For co-localization experiments, αSYN phase separation was induced in the presence of 2 mM Ca²⁺ and 15% PEG 8000. αSYN was supplemented with 1% Alexa 488 labelled αSYN. VAMP1-96 and syntaxin1-265, supplemented with 1% Alexa 594 labelled protein, were added after droplet formation and imaged after 30 min of incubation. Data were obtained using the same imaging conditions for VAMP1-96 and syntaxin1-256, respectively. Images analysis was performed in Fiji[122]. In brief, a mask for αSYN droplets was generated in the 488 channel, which was used to measure the intensity of αSYN in the 488 channel and the intensity of VAMP1-96 and syntaxin1-265 in the 594 channel. The ratio of VAMP1-96 and syntaxin1-265 to αSYN intensity was plotted. Data from at least three independent sets of biological samples were obtained. For sedimentation-based assays, αSYN phase separation was

induced in the presence of 2 mM Ca$^{2+}$ and 15% PEG 8000, followed by the addition of VAMP1-96 protein. Samples of 50 µl reaction volume were set up and incubated for 20 min. Samples were centrifuged at 17,115$g$ for 20 min at 25 °C to separate the dense phase (pellet fraction) and the light phase (supernatant). The supernatant was carefully removed, and the pellet was resuspended in 8 M urea. The samples were run on a 15% SDS–PAGE gel and were visualized using Coomassie blue staining (Quick Coomassie stain, Protein Ark). The gels were scanned on a ChemiDoc MP Imaging System (Bio-Rad) using ImageLab v.6.0.1. $C_{sat}$ was calculated from the band intensity referenced to a known αSYN standard using Fiji software[122,123].

## NMR spectroscopy

Recombinant $^{15}$N-labelled wild-type human full-length αSYN was grown in M9 minimal medium supplemented with a vitamin mix, trace elements and the appropriate nitrogen ($^{15}$NH$_4$) and carbon ($^{12}$C-glucose) sources and purified as described above. The $^{15}$N-labelled αSYN was dialysed into 20 mM phosphate buffer, pH 7.0. Samples of 200 µl were prepared for NMR spectroscopy containing 40 µM αSYN, the respective concentration of peptide (custom synthesis by Proteogenix, Schiltigheim) and 10% $^2$H$_2$O (Thermo Scientific Chemicals). To probe the interaction of peptide with αSYN at a residue specific level, we used multi-dimensional NMR spectroscopy and recorded $^1$H-$^{15}$N-BEST-TROSY spectra[79,80] titrating the concentration of the JMD long peptide (50, 100 and 150 µM peptide) at a fixed concentration of αSYN of 40 µM. The inter-scan delay was set to 0.3 s. NMR experiments were carried out at 800 MHz and 600 MHz using Bruker Avance spectrometers equipped with 5-mm triple resonance inverse cryoprobes, at 298 K. Assignments of the resonances in the $^1$H-$^{15}$N-BEST-TROSY spectra of αSYN were derived from previous studies[87]. Spectra were internally referenced to the $^1$H$_2$O signal at 4.70 ppm, processed with TopSpin v.2.1 (Bruker) and analysed with SPARKY[124].

## Plasmids

Wild-type human full-length *SNCA* and *VAMP2*, encoding αSYN and VAMP2, were cloned from complementary DNA obtained from human neuroblastoma cells (SH-SY5Y) and inserted into the pEYFP-N1 and pMD2.G vector (Addgene 96808 and 12259) with a C-terminal YFP and Flag-tag, respectively. Synaptojanin 145 and 170, endophilin-A1 and HSC70 constructs were purchased from Addgene (22291, 22292, 47403 and 86031), pcDNA3-Flag-synaptojanin 1-145 and synaptojanin 1-170 were a gift from P. De Camilli (Addgene plasmid 22291; RRID: Addgene_22291; Addgene plasmid 22292; RRID: Addgene_22292)[125], full-length endophilin was a gift from P. McPherson (Addgene plasmid 47403; RRID: Addgene_47403)[126], pCDNAZeo(-)HSC73 AS was a gift from J. Blum (Addgene plasmid 86031; RRID: Addgene_86031)[127]. All other synaptic genes, *SNCAIP*, *RAB3A*, *RPH3A*, *VPS35*, *DNAJC6*, *STX1A* and *SNAP25*, encoding synphilin-1, Rab-3A, rabphilin-3A, VPS35, auxilin, syntaxin-1A and SNAP25, were cloned from SH-SY5Y cDNA and were then inserted into the pMD2.G vector. *SYT1*, encoding synaptotagmin-1, was cloned from SH-SY5Y cDNA into the pCDNA3.1 vector with an N-terminal mScarlet tag, *CPLX1* and *CLPX2*, encoding complexin-1 and complexin-2 were cloned from SH-SY5Y cDNA into the pCDNA3.1 vector with a C-terminal mScarlet tag (Addgene 16015 and Addgene 85045). Gibson assembly was performed upon PCR amplification (Q5 Hot start HiFi 2xMM, M0494S; 2xHiFi DNA Assembly MM, E2621S, NEB). αSYN(96AAA) and αSYN(A30)P were generated using KLD substitution (M0554S, NEB). All sequences were verified by sequencing. Primer sequences are provided as Supplementary Data. 5HT6-YFP-Inpp5e was a gift from T. Inoue (Addgene plasmid 96808; RRID: Addgene_96808)[128]. pMD2.G was a gift from D.Trono (Addgene plasmid 12259; RRID: Addgene_12259). pcDNA3.1/V5-His Snk/Plk2 was a gift from W. El-Deiry (Addgene plasmid 16015; RRID: Addgene_16015)[129]. pmScarlet_Giantin_C1 was a gift from D. Gadella (Addgene plasmid 85048; RRID: Addgene_85048)[130].

## Cell culture and transfection

HeLa cells were obtained from the European Collection of Cell Cultures (ECACC 93021013) and grown in high-glucose DMEM (31966-021, Gibco) supplemented with 10% FBS (F7524, Sigma) and 1% penicillin/streptomycin (P0781, Sigma). Cells were grown at 37 °C in a humidified incubator with 5% CO$_2$. Cells were tested for *Mycoplasma* contamination using MycoStrip (IvivoGen). Cells were plated at 20,000 cells per well in 48-well plates (Cellstar, 677 180, Greiner Bio-one) for Incucyte experiments or in eight-well Ibidi dishes (80807, Ibidi) for confocal imaging. Cells were transfected the following day using Fugene HD Transfection reagent according to the manufacturer's protocol (E2311, Promega). In brief, per reaction 12.5 µl OptiMEM (31985-062, Gibco) was set up in 1.5-ml sterile Eppendorf tubes. A total of 250 ng DNA and 0.75 µl Fugene reagent were added and incubated for 15 min at room temperature. The transfection mix was added to the cells for 1 min and then topped up with 300 µl complete medium. Cells were imaged the next day.

## Incucyte, confocal microscopy and image analysis

Cells in 48-well plates were imaged with the Incucyte S3 (Essen BioScience). Phase brightfield and green fluorescence images were taken using a 20× objective at a 4-h interval at 200 ms exposure, condensate formation (% of cells showing condensate formation) was evaluated 16 h after transfection. Incucyte 2021A was used for data analysis. At least three biological repeats with three technical repeats each were analysed blinded to the investigator. Live cell confocal imaging was performed on an LSM780 microscope (Zeiss) using a 63× oil immersion objective if not stated otherwise. YFP fluorescence was excited with the 514 laser at 2% laser power, mScarlet was excited using the 561 laser at 2% laser intensity. Zen 2.3 (black edition) and Zen 2.6 (blue edition) were used for data collection. Fluid-like behaviour of αSYN condensates was demonstrated by time-course imaging coupled with z-stacks using the Super-resolution (SR) Airyscan feature of the LSM 880 microscope (Zeiss), followed by 3D reconstruction using Imaris v.10.1.0 software. Cells were imaged at ×8 zoom, 372 × 372-pixel resolution over a range of 10–14 z-planes and for 8–10 cycles using a 63× oil immersion objective. Images were reconstructed to generate 3D surface-masked movies of αSYN condensates. For FRAP experiments images were taken with the 63× oil immersion objective, 20× zoom, 128 × 128-pixel resolution, at an imaging speed of 60 ms per image. Three pre-bleach images were acquired before the region of interest was bleached with 100 iterations at 100% laser power using the 514 laser. Fluorescence recovery was recorded for 100 cycles. FRAP analysis was performed in Fiji using the FRAP profiler v.2 plugin (Hardin laboratory, https://worms.zoology.wisc.edu/research/4d/4d.html). Small aliphatic alcohols were used to evaluate the role of hydrophobic interaction for αSYN condensate formation. 1,6-hexanediol (240117, Sigma) and 1,3-propanediol (P50404, Sigma) were prepared as 6% stock solutions in complete DMEM and were added to the cells in a 1:1 ratio after the first image was taken. Condensate size, number of condensates per cell and the ratio of condensate intensity and αSYN cytosolic intensity for wild-type αSYN and αSYN(96AAA) were analysed using Fiji[122]. Analysis was performed blinded to the investigator. Pearson correlation coefficients were calculated using the ColocFinder Plugin (https://imagej.nih.gov/ij/plugins/colocalization-finder.html). Profile plots were generated in Fiji[122] and plotted using GraphPad Prism v.9.3.1.

## Immunocytochemistry

HeLa cells were plated on eight-well Ibidi dishes (80807, Ibidi), transfected as above and fixed the following day using 4% paraformaldehyde in phosphate-buffered saline (PBS), pH 7.4. Blocking and permeabilization were performed using 10% FBS, 1% BSA and 0.3% Triton X-100 in PBS for 1 h. Cells were stained using an anti-Flag primary antibody raised in rabbit (20543-1-AP, Proteintech), used at 1:1,000 dilution in PBS containing 1% BSA and incubated overnight at 4 °C. Following three washes with PBS, secondary antibody (Alexa Fluor 594, A11072 or Alexa

Fluor 647, A21246, Invitrogen) diluted at 1:1,000 in PBS with 1% BSA was added to the cells and incubated at room temperature for 1 h. After three washes with PBS, cells were imaged on an LSM780 microscope (Zeiss) using a 20× air or 40× oil immersion objective. YFP fluorescence was excited with the 514 nm laser, Flag staining was detected using the 561 nm or the 633 nm laser depending on the secondary antibody used. The total number of αSYN–YFP-expressing cells and the number of cells expressing both αSYN–YFP and VAMP2–Flag were counted. Number of cells positive for αSYN–YFP and VAMP2 were expressed as the percentage of all αSYN–YFP-positive cells.

## Correlative light and electron microscopy

HeLa cells were plated on gridded Mattek dishes (Ashland) at a density of 20,000 cells per dish and transfected as above with plasmids encoding αSYN–YFP, VAMP2 and 4xMTS-mScarlet[131], to visualize mitochondria relative to the location of αSYN condensates. 4xmts-mScarlet-I was a gift from D. Gadella (Addgene plasmid 98818; RRID: Addgene_98818). The following day cells were fixed using 2% paraformaldehyde/2.5% glutaraldehyde in 0.1 M Na cacodylate buffer, pH 7.2. Cells with condensates were imaged on an LSM780 confocal microscope (Zeiss) using 63× oil immersion objectives. The location of the cells was noted based on the etched numeric grid to allow subsequent analysis of the same cells by electron microscopy. Cells were then washed with 0.1 M Na cacodylate buffer, pH 7.2, post-fixed in 1% osmium tetroxide in 0.1 M Na cacodylate buffer, pH 7.2, for 1 h and washed with 0.1 M Na cacodylate buffer, pH 7.2 and 0.05 M sodium maleate buffer, pH 5.2. The cells were stained en bloc with 0.5% uranyl acetate in 0.05 M sodium maleate buffer in the dark for 1 h, washed with 0.05 M maleate buffer, pH 5.2 and dehydrated sequentially through 50%, 70%, 90% and 2 × 100% ethanol for 10 min each. Infiltration of the sample was accomplished with 50:50 ethanol:Agar 100 resin (20 min) and subsequently with pure Agar 100 resin (Agar Scientific) for 24 h. BEEM capsules were filled with resin and inverted over the cell monolayer before polymerization of the resin overnight at 60 °C to embed the cells 'en face'. The BEEM capsules and exposed monolayer of cells were removed from the culture dishes by immersion in liquid nitrogen.

Ultrathin sections were cut 'en face' to the plane of the monolayer using a diamond knife mounted on a Leica Ultracut UC7 ultramicrotome (Leica), collected on slot electron microscopy grids and stained with uranyl acetate and Reynold's lead citrate. The sections were observed in a Tecnai G2 Spirit BIOTWIN transmission electron microscope (FEI) at an operating voltage of 80 kV and images were recorded using a 4-megapixel Gatan US1000 CCD camera.

## Quantification and statistical analysis

Data analysis and statistical analysis was performed using Excel 2016 and GraphPad Prism v.9.3.1. All data are represented as mean ± s.d. Statistical analysis was carried out using unpaired two-tailed *t*-test or one-way ANOVA with Dunnett's multiple comparison test. For 1,6-hexanediol and 1,3-propanediol experiments a paired two-tailed *t*-test and repeated measures one-way ANOVA with Geisser-Greenhouse correction and Dunnett's multiple comparisons test were performed, respectively. Statistical parameters are reported in the figures and the corresponding legends. Exact *P* values are shown throughout the manuscript. Data distribution was assumed to be normal but this was not formally tested. No statistical methods were used to predetermine sample sizes but our sample sizes are similar to those reported in previous publications[11,91,92]. Samples were randomly allocated into experimental groups. Data collection and analysis have been performed blinded when indicated. Data were included if the control (WT) showed appropriate condensate formation.

## Reporting summary

Further information on research design is available in the Nature Portfolio Reporting Summary linked to this article.

## Data availability

NMR data have been deposited to BMRBdep (https://bmrb.io/deposit/) under deposition numbers 52093, 52094 and 52095. Plasmids generated in this study are available from the corresponding author with a completed material transfer agreement. All other data supporting the findings of this study are available from the corresponding author on reasonable request. Source data are provided with this paper.

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

## Acknowledgements

J.L. acknowledges funding from the Royal Society (Royal Society Dorothy Hodgkin Research Fellowship, DHF/R1/201228), the Addenbrooke's Charitable Trust (grant award 900325), the Leverhulme Trust (research project grant, RPG-2022-257) and a Career Support Fund from the University of Cambridge. We thank M. G. Martin and J. L. Gonzalez for electron microscopy support. A.J.W. acknowledges funding from Blood Cancer UK (21002), the UK Medical Research Council (MR/T012412/1) and the Rosetrees Trust (PGL22/100032). Furthermore, we thank D. Gershlick and C. Lautenschläger for discussion on the manuscript.

## Author contributions

Conceptualization was the responsibility of J.L. Methodology was the responsibility of A.A., A.C., C.H., N.A.B., N.M. and J.L. Investigation was the responsibility of A.A., A.C., F.R., I.M.U., C.H., K.S., N.A.B., N.M. and

J.L. Writing of the original draft was conducted by J.L. and A.A. Review and editing was carried out by J.L., A.A., A.C., C.H. and A.J.W. Funding acquisition was the responsibility of A.J.W. and J.L. Supervision was carried out by A.J.W. and J.L.

## Competing interests

The authors declare no competing interests.

## Additional information

**Extended data** is available for this paper at https://doi.org/10.1038/s41556-024-01451-6.

**Correspondence and requests for materials** should be addressed to Janin Lautenschläger.

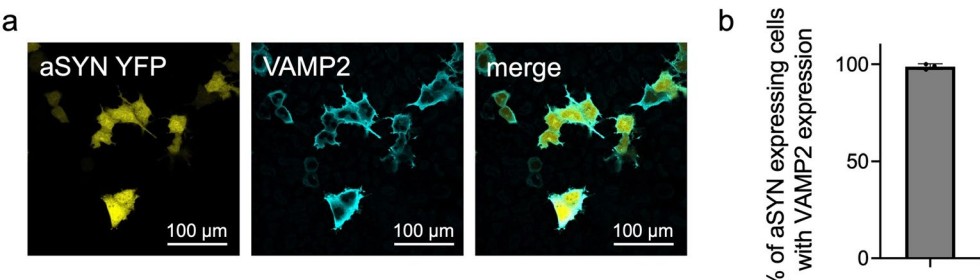

**Extended Data Fig. 1 | αSYN YFP / VAMP2 co-expression. a)** Cells expressing aSYN YFP also show expression of VAMP2 as revealed upon immunocytochemistry. **b)** Quantification of aSYN YFP expressing cells and cells expressing aSYN YFP and immunostained for VAMP2-Flag shows that between 97 to 100% of aSYN YFP cells show VAMP2 co-expression. n = 22 images, pooled from 3 biological repeats. Data are represented as mean ± s.d..

**Extended Data Fig. 2 | Non-rendered images of aSYN condensate fusion and fission.** Non-rendered image showing aSYN YFP condensates displayed in Fig. 2e (full frame, left image). Right panels showing zoom in and time frames for aSYN YFP condensate fusion (upper panel) and aSYN YFP condensate fission (lower panel). This experiment has been repeated independently two times with similar results. Scale bars in zoom in images indicates 1 μm.

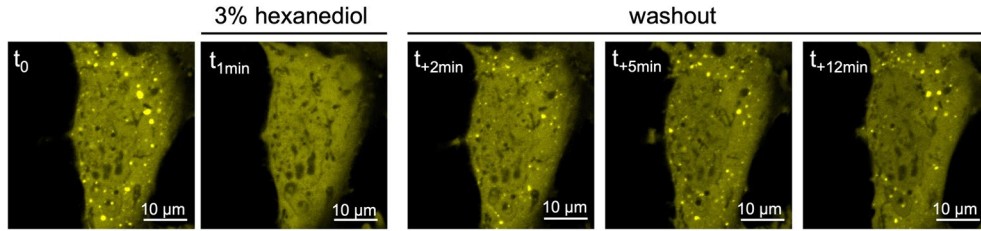

**Extended Data Fig. 3 | Sensitivity of alpha-synuclein condensates to 1,6-hexanediol.** aSYN YFP condensates show dispersal upon incubation with 3% 1,6-hexanediol with fast recovery after 1,6-hexanediol washout. This experiment has been repeated independently three times with similar results.

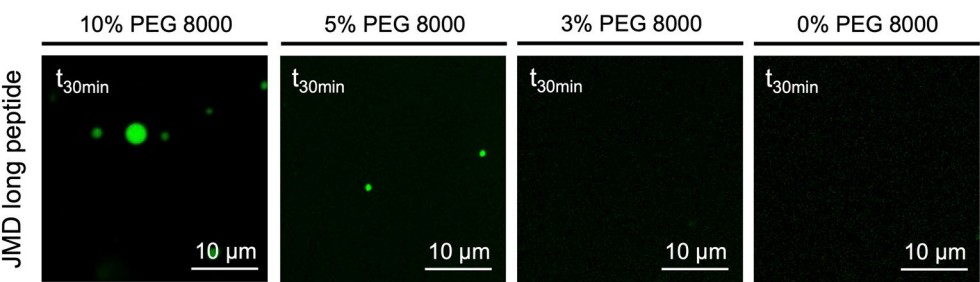

**Extended Data Fig. 4 | aSYN condensate formation at lower PEG concentrations.** Evaluation of aSYN phase separation in the presence of 150 µM peptide and crowding with 10%, 5%, 3% and 0% PEG 8000 respectively. 30 min after PEG addition ($t_{30min}$), 40 µM aSYN. This experiment has been repeated independently twice with similar results.

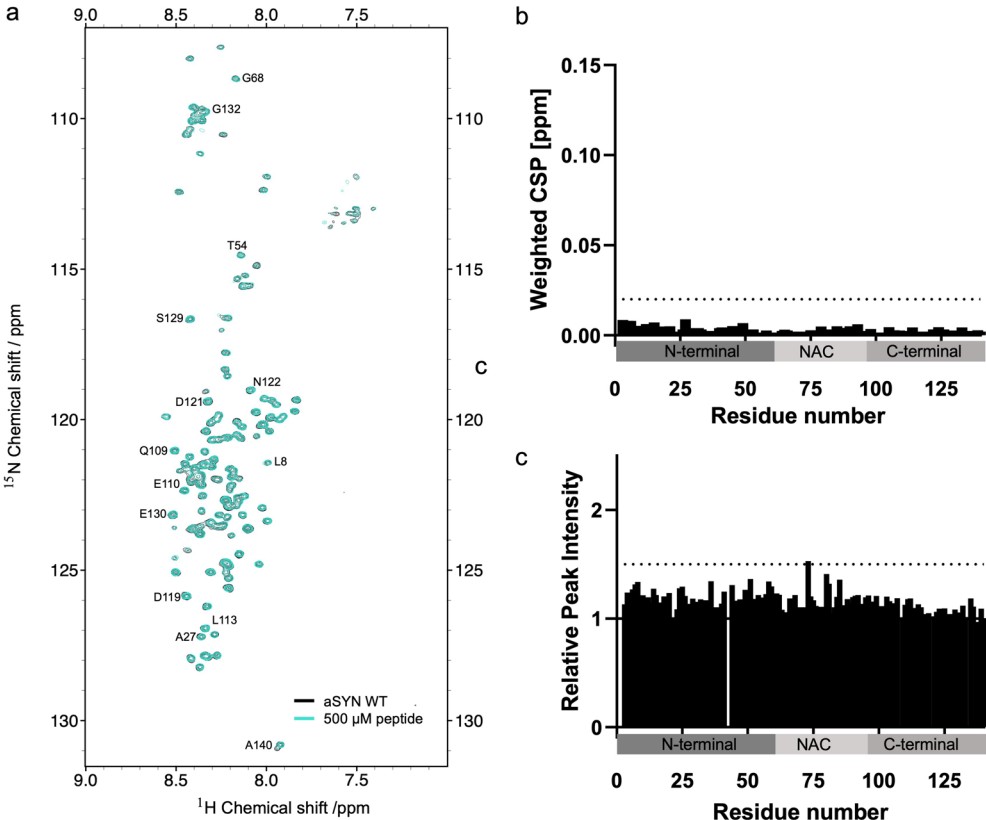

**Extended Data Fig. 5 | 1H-15N-BEST-TROSY spectra of aSYN in the presence of NT long peptide 1. a**) Overlapped $^1$H-$^{15}$N-BEST-TROSY spectra of aSYN without (black) and in the presence (turquoise) of NT long peptide 1. aSYN concentration used: 40 µM, peptide concentration: 500 µM. **b**) Weighted average (of $^{15}$N and $^1$H) chemical shift perturbation ($\Delta\delta = \sqrt{(\delta H^2 + 0.2\,(\delta^{15}N)^2)}$) of residues in aSYN in the presence of 500 µM NT long peptide 1. Dashed line indicates Weighted CSP of 0.02 ppm. **c**) Relative peak intensities of aSYN residues in the presence of 500 µM NT long peptide 1. Black dashed line indicates Relative Peak Intensity of 1.5.

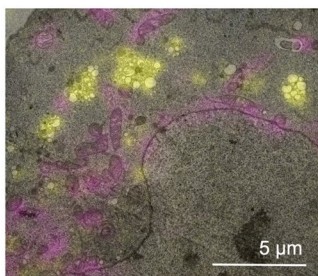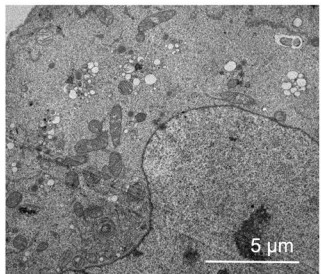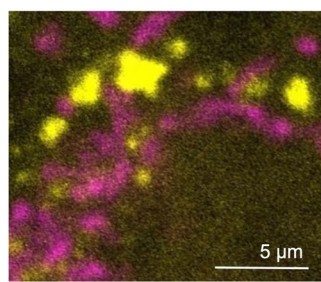

**Extended Data Fig. 6 | Correlative Light and Electron Microscopy (CLEM), Supplement for Fig. 7g.** Light microscopy and electron microscopy overlay, original electron microscopy and light microscopy image side by side. This experiment has been repeated independently twice with similar results.

# Reporting Summary

## Statistics

For all statistical analyses, confirm that the following items are present in the figure legend, table legend, main text, or Methods section.

| n/a | Confirmed | |
|---|---|---|
| ☐ | ☒ | The exact sample size (*n*) for each experimental group/condition, given as a discrete number and unit of measurement |
| ☐ | ☒ | A statement on whether measurements were taken from distinct samples or whether the same sample was measured repeatedly |
| ☐ | ☒ | The statistical test(s) used AND whether they are one- or two-sided |
| | | *Only common tests should be described solely by name; describe more complex techniques in the Methods section.* |
| ☒ | ☐ | A description of all covariates tested |
| ☐ | ☒ | A description of any assumptions or corrections, such as tests of normality and adjustment for multiple comparisons |
| ☐ | ☒ | A full description of the statistical parameters including central tendency (e.g. means) or other basic estimates (e.g. regression coefficient) AND variation (e.g. standard deviation) or associated estimates of uncertainty (e.g. confidence intervals) |
| ☐ | ☒ | For null hypothesis testing, the test statistic (e.g. *F*, *t*, *r*) with confidence intervals, effect sizes, degrees of freedom and *P* value noted |
| | | *Give P values as exact values whenever suitable.* |
| ☒ | ☐ | For Bayesian analysis, information on the choice of priors and Markov chain Monte Carlo settings |
| ☒ | ☐ | For hierarchical and complex designs, identification of the appropriate level for tests and full reporting of outcomes |
| ☐ | ☒ | Estimates of effect sizes (e.g. Cohen's *d*, Pearson's *r*), indicating how they were calculated |

*Our web collection on statistics for biologists contains articles on many of the points above.*

## Software and code

Policy information about availability of computer code

| Data collection | Data collection was performed using Zen 2.3 (black edition) and Zen 2.6 (blue edition), IncuCyte 2021A, CLARIOStar 5.01 and ImageLab 6.0.1 |
|---|---|
| Data analysis | Image analysis was performed using FIJI and Imaris 10.1.0, data analysis was performed using Excel 2016, OriginPro 2018, and Prism 9.3.1 |

For manuscripts utilizing custom algorithms or software that are central to the research but not yet described in published literature, software must be made available to editors and reviewers. We strongly encourage code deposition in a community repository (e.g. GitHub). See the Nature Portfolio guidelines for submitting code & software for further information.

## Data

Policy information about availability of data

All manuscripts must include a data availability statement. This statement should provide the following information, where applicable:
- Accession codes, unique identifiers, or web links for publicly available datasets
- A description of any restrictions on data availability
- For clinical datasets or third party data, please ensure that the statement adheres to our policy

All Source data are provided with the study. NMR data have been deposited to BMRBdep (https://bmrb.io/deposit/) under the deposition number 52093, 52094, and 52095.

## Research involving human participants, their data, or biological material

Policy information about studies with [human participants or human data](). See also policy information about [sex, gender (identity/presentation), and sexual orientation]() and [race, ethnicity and racism]().

| | |
|---|---|
| Reporting on sex and gender | N/A |
| Reporting on race, ethnicity, or other socially relevant groupings | N/A |
| Population characteristics | N/A |
| Recruitment | N/A |
| Ethics oversight | N/A |

Note that full information on the approval of the study protocol must also be provided in the manuscript.

# Field-specific reporting

Please select the one below that is the best fit for your research. If you are not sure, read the appropriate sections before making your selection.

☒ Life sciences        ☐ Behavioural & social sciences        ☐ Ecological, evolutionary & environmental sciences

For a reference copy of the document with all sections, see [nature.com/documents/nr-reporting-summary-flat.pdf](nature.com/documents/nr-reporting-summary-flat.pdf)

# Life sciences study design

All studies must disclose on these points even when the disclosure is negative.

| | |
|---|---|
| Sample size | No statistical methods were used to pre-determine sample sizes but our sample sizes are similar to those reported in previous publications (Wu et al. 2021 Mol. Cell; Park et al. 2021 Nat. Commun; Park et al. 2023 Nat. Commun). |
| Data exclusions | Data were included if the control (wildtype) showed appropriate condensate formation. |
| Replication | All experiments have been replicated 2 or more times as indicated in the manuscript. All attempts of replication were successful. |
| Randomization | Samples were randomly allocated into experimental groups. |
| Blinding | Blinded analysis was performed for quantification of condensate formation (Incucyte experiments) and for quantitative analysis of condensates of the 96AAA aSYN variant. Image acquisition and analysis were performed non-blinded otherwise, but performed to an objectively defined standard and ensuring reproducibility. |

# Reporting for specific materials, systems and methods

We require information from authors about some types of materials, experimental systems and methods used in many studies. Here, indicate whether each material, system or method listed is relevant to your study. If you are not sure if a list item applies to your research, read the appropriate section before selecting a response.

### Materials & experimental systems

| n/a | Involved in the study |
|---|---|
| ☐ | ☒ Antibodies |
| ☐ | ☒ Eukaryotic cell lines |
| ☒ | ☐ Palaeontology and archaeology |
| ☒ | ☐ Animals and other organisms |
| ☒ | ☐ Clinical data |
| ☒ | ☐ Dual use research of concern |
| ☒ | ☐ Plants |

### Methods

| n/a | Involved in the study |
|---|---|
| ☒ | ☐ ChIP-seq |
| ☒ | ☐ Flow cytometry |
| ☒ | ☐ MRI-based neuroimaging |

## Antibodies

| | |
|---|---|
| Antibodies used | Anti-Flag Rabbit PolyAb (Proteintech, Cat No. 20543-1-AP, Lot 00098866) |

| Validation | Anti-Flag Rabbit PolyAb (Proteintech, Cat No. 20543-1-AP, Lot 00098866)

Manufacturer validation: HEK-293T cells and transfected HEK-293T lysates were subjected to SDS PAGE followed by western blot with 20543-1-AP (DYKDDDDK tag antibody) at dilution of 1:50000 incubated at room temperature for 1.5 hours. No band is seen for un-transfected HEK-293T cells. Western blot image is provided on the manufacturer website: https://www.ptglab.com/products/Flag-Tag-Antibody-20543-1-AP.htm

20543-1-AP targets DYKDDDDK tag in WB, IP, IF, FC, RIP, IHC, CoIP, ChIP, ELISA applications and shows reactivity with recombinant protein samples. Cited reactivity: Human, Mouse, Rat, Chicken, Yeast, Monkey, Pig, Duck, Immunogen: DYKDDDDK tag fusion protein Ag2329. RRID: AB_11232216 |

## Eukaryotic cell lines

Policy information about cell lines and Sex and Gender in Research

| Cell line source(s) | HeLa cells were obtained from the European Collection of Cell Cultures (ECACC 93021013). sex: female, human cell line |
|---|---|
| Authentication | Cell morphology |
| Mycoplasma contamination | Cells tested negative for mycoplasma contamination using the MycoStripTM assay (InvivoGen). |
| Commonly misidentified lines (See ICLAC register) | No commonly misidentified lines per the ICLAC register were used. |

