## [Peer Review File · Nature Cell Biology]

Peer Review Information

Journal: Nature Cell Biology

Manuscript Title: VAMP2 regulates phase separation of α -synuclein

Corresponding author name(s): Dr Janin Lautenschläger

Editorial Notes:

Reviewer Comments & Decisions:

Decision Letter, initial version:
--

*Please delete the link to your author homepage if you wish to forward this email to co-authors.

Dear Dr Lautenschläger,

I am truly sorry for the very long delay in the peer review of your submission. Despite our best efforts, we have been unable to receive the comments of reviewer #3, who is an expert in alpha-Synuclein. If/when we do receive the missing comments, we will pass them on to you. I do apologize for the delayed decision process.

Your manuscript, "VAMP2 regulates phase separation of alpha-synuclein", has now been seen by 2 referees, who are experts in NMR and biomolecular condensates (referee 1); SNAREs (referee 2). As you will see from their comments (attached below) they find this work of potential interest, but have raised substantial concerns, which in our view would need to be addressed with considerable revisions before we can consider publication in Nature Cell Biology.

Nature Cell Biology editors discuss the referee reports in detail within the editorial team, including the chief editor, to identify key referee points that should be addressed with priority, and requests that are overruled as being beyond the scope of the current study. To guide the scope of the revisions, I have listed these points below. We are committed to providing a fair and constructive peer-review process, so please feel free to contact me if you would like to discuss any of the referee comments further.

In particular, it would be essential to:

A) Assess potential effects of disrupting alpha-Synuclein – VAMP2 interactions on neurotransmitter release (Reviewer #2)

B) Experimentally test claims on biomolecular condensation, especially in regard to effects from 1,6-HD perturbations (Reviewer #1)

C) Clarify NMR datasets, particularly with respect to any potential additional interactions, as well as whether such changes in proposed interactions have effects on biomolecular condensation (reviewer #1)

D) All other referee concerns pertaining to strengthening existing data, providing controls, methodological details, clarifications and textual changes, should also be addressed.

E) Finally please pay close attention to our guidelines on statistical and methodological reporting (listed below) as failure to do so may delay the reconsideration of the revised manuscript. In particular please provide:

We would be happy to consider a revised manuscript that would satisfactorily address these points, unless a similar paper is published elsewhere, or is accepted for publication in Nature Cell Biology in the meantime.

- ensure that it conforms to our format instructions and publication policies (see below and www.nature.com/nature/authors/).

- provide a point-by-point rebuttal to the full referee reports verbatim, as provided at the end of this letter.

- provide the completed Editorial Policy Checklist (found here <https://www.nature.com/authors/policies/Policy.pdf>), and Reporting Summary (found here <https://www.nature.com/authors/policies/ReportingSummary.pdf>). This is essential for reconsideration of the manuscript and these documents will be available to editors and referees in the event of peer review. For more information see <http://www.nature.com/authors/policies/availability.html> or contact me.

Nature Cell Biology is committed to improving transparency in authorship. As part of our efforts in this direction, we are now requesting that all authors identified as 'corresponding author' on published papers create and link their Open Researcher and Contributor Identifier (ORCID) with their account on the Manuscript Tracking System (MTS), prior to acceptance. ORCID helps the scientific community

achieve unambiguous attribution of all scholarly contributions. You can create and link your ORCID from the home page of the MTS by clicking on 'Modify my Springer Nature account'. For more information please visit www.springernature.com/orcid.

[Redacted]

We would like to receive a revised submission within six months. We would be happy to consider a revision even after this timeframe, however if the resubmission deadline is missed and the paper is eventually published, the submission date will be the date when the revised manuscript was received.

We hope that you will find our referees' comments, and editorial guidance helpful. Please do not hesitate to contact me if there is anything you would like to discuss.

Best wishes,

Daryl

Daryl Jason Verzosa David, PhD

Senior Editor, Nature Cell Biology
Nature Portfolio

Heidelberger Platz 3, 14197 Berlin, Germany
Email: daryl.david@nature.com
ORCID: <https://orcid.org/0000-0002-9253-4805>

Reviewers' Comments:

Reviewer #1:

Remarks to the Author:

In this manuscript the authors demonstrate that aSYN phase separation is promoted by an interaction with a positively-charged region near the juxtamembrane region of VAMP2 and dependent on aSYN's capacity to bind to lipid membranes. Notably, co-expression of VAMP2 and aSYN induces the formation of dynamic aSYN-containing condensates in cells. The authors also demonstrate that aSYN condensates accumulate in vesicular structures in cells. Disruption of aSYN lipid binding by the pathological mutation A30P also disrupts condensate formation indicating a possible link between aSYN condensation, SNARE function and disease. The conclusions of the paper are original. It is well-

written and conclusions are supported by the presented data, using a range of approaches in vitro and in cells, all of which appear to be rigorously performed with appropriate statistical analysis.

We make the following suggestions:

Please define aSYN 96AAA at first usage.

More useful information should be extracted from the NMR data. For example, the aSYN spectra (Fig 6) demonstrate that the protein is disordered. The clustering of the shifts and the direction suggest a potential stabilization of a helix. Were the intensities of the peaks compared for the datasets? Chemical shifts are not necessarily correlated with interaction strength. Do the intensity changes upon binding reveal any additional interactions between the peptide and the aSYN? In addition, the NMR experiment "1H-15N-BEST-TROSY" should be referenced.

"However, when aSYN YFP was co-expressed with VAMP2, cytosolic aSYN clusters were observed in about 5-10% of the transfected cells" – Is this because the two proteins are only over-expressed in 10% of the cells? Can the authors determine if VAMP2 is not being expressed in cells that don't have the puncta? More information is required on this key point.

Although 1,6-hexanediol is often used to disrupt condensates, it is not a general condensate dissolver. It seems an unlikely candidate for disrupting electrostatic interactions. It suggests that electrostatic, polar/H-bonding and hydrophobic interactions contribute to condensate formation in this case. Alternatively, this 1,6-hexanediol effect might be non-specific.

"To test the relevance of lipid binding we used aSYN A30P, an aSYN disease variant 79 with defective lipid binding 31,38,80–82 little effect of A30P." – This sentence is unclear.

"This is interesting since immunoprecipitation experiments show that the N-terminal proline-rich domain of VAMP2 mediates VAMP2/aSYN interaction" -- The data do not indicate that the N-terminal proline rich region does not contribute to phase separation within the context of the full protein, only that the relatively small peptides used do not promote phase separation. With respect to figure 5, it should be noted that short peptides, which are important for phase separation in the context of the protein, can actually impede phase separation when extracted from the larger protein by acting as "caps". If they have a valence of 1, they can block interactions that contribute to multivalency (see Sanders, D., Kedersha, N. et al. Cell 2020, 181:2 for an explanation). This is another reason to analyze the intensity change in the NMR spectra as they will provide information on interaction not visible in the chemical shift perturbations. For disordered proteins, chemical shift perturbations often rely on large chemical shift changes effected by changes in secondary structure as they are typically in the fast exchange regime. In fact, for many disordered protein interactions, no chemical shift changes are observed, though intensity changes can be dramatic.

"LLPS" should not be used as a keyword since it is not used in the manuscript and the data are not definitive that this is simple LLPS. Many biomolecular condensates use more complex phase separation processes (see DOI: 10.1016/j.molcel.2022.05.018), and the manuscript, appropriately, refers only to "phase separation". Thus, the keyword should be "phase separation".

Reviewer #2:

Remarks to the Author:

The authors co-express alpha-synuclein (aSyn) with a panel of Parkinson's Disease associated proteins in HeLa cells and find that one of them, VAMP2, leads to the formation of phase-separated condensates. In vitro, VAMP2 promotes liquid-liquid phase separation (LLPS) of aSyn, with which it forms co-condensates. A 14-residue peptide representing the juxtamembrane region of VAMP2 strongly enhanced aSyn LLPS, even in the absence of non-physiologically high levels of calcium which was otherwise required. NMR experiments demonstrate that this VAMP2 peptide primarily interacts with the C-terminal portion of aSyn. In vivo, LLPS appears to depend on the membrane association of aSyn, as it is abolished by an aSyn mutant (A30P) with reduced membrane binding. EM reveals cytosolic accumulation of heterogeneously sized vesicles in cells expressing both aSyn and VAMP2, consistent with the idea that aSyn and VAMP2 form membrane associated condensates.

These results are interesting and they do, in my judgment, advance the field. My main concern is whether we learn enough about the relationship between aSyn:VAMP2 LLPS and synaptic vesicle cycling to justify publication in Nature Cell Biology. As suggested by the authors, the link between aSyn phase separation and VAMP2 is intriguing and hints at a possible regulatory mechanism linked to SNARE assembly/disassembly. It will be interesting to test this hypothesis by taking advantage of the structural insights gained in this work.

Reviewer #3:

None

Methods should be written concisely, but should contain all elements necessary to allow interpretation and replication of the results. As a guideline, Methods sections typically do not exceed 3,000 words. The Methods should be divided into subsections listing reagents and techniques. When citing previous methods, accurate references should be provided and any alterations should be noted. Information must be provided about: antibody dilutions, company names, catalogue numbers and clone numbers for monoclonal antibodies; sequences of RNAi and cDNA probes/primers or company names and

catalogue numbers if reagents are commercial; cell line names, sources and information on cell line identity and authentication. Animal studies and experiments involving human subjects must be reported in detail, identifying the committees approving the protocols. For studies involving human subjects/samples, a statement must be included confirming that informed consent was obtained. Statistical analyses and information on the reproducibility of experimental results should be provided in a section titled "Statistics and Reproducibility".

All Nature Cell Biology manuscripts submitted on or after March 21 2016 must include a Data availability statement at the end of the Methods section. For Springer Nature policies on data availability see <http://www.nature.com/authors/policies/availability.html>; for more information on this particular policy see <http://www.nature.com/authors/policies/data/data-availability-statements-data-citations.pdf>. The Data availability statement should include:

- Accession codes for primary datasets (generated during the study under consideration and designated as "primary accessions") and secondary datasets (published datasets reanalysed during the study under consideration, designated as "referenced accessions"). For primary accessions data should be made public to coincide with publication of the manuscript. A list of data types for which submission to community-endorsed public repositories is mandated (including sequence, structure, microarray, deep sequencing data) can be found here <http://www.nature.com/authors/policies/availability.html#data>.
- Unique identifiers (accession codes, DOIs or other unique persistent identifier) and hyperlinks for datasets deposited in an approved repository, but for which data deposition is not mandated (see here for details <http://www.nature.com/sdata/data-policies/repositories>).
- At a minimum, please include a statement confirming that all relevant data are available from the authors, and/or are included with the manuscript (e.g. as source data or supplementary information), listing which data are included (e.g. by figure panels and data types) and mentioning any restrictions on availability.
- If a dataset has a Digital Object Identifier (DOI) as its unique identifier, we strongly encourage including this in the Reference list and citing the dataset in the Methods.

We recommend that you upload the step-by-step protocols used in this manuscript to the Protocol Exchange. More details can found at www.nature.com/protocolexchange/about.

All imaging data should be accompanied by scale bars, which should be defined in the legend. Cropped images of gels/blots are acceptable, but need to be accompanied by size markers, and to

retain visible background signal within the linear range (i.e. should not be saturated). The boundaries of panels with low background have to be demarked with black lines. Splicing of panels should only be considered if unavoidable, and must be clearly marked on the figure, and noted in the legend with a statement on whether the samples were obtained and processed simultaneously. Quantitative comparisons between samples on different gels/blots are discouraged; if this is unavoidable, it should only be performed for samples derived from the same experiment with gels/blots were processed in parallel, which needs to be stated in the legend.

- For line art, graphs, charts and schematics we prefer Adobe Illustrator (.AI), Encapsulated PostScript (.EPS) or Portable Document Format (.PDF). Files should be saved or exported as such directly from the application in which they were made, to allow us to restyle them according to our journal house style.
- We accept PowerPoint (.PPT) files if they are fully editable. However, please refrain from adding PowerPoint graphical effects to objects, as this results in them outputting poor quality raster art. Text used for PowerPoint figures should be Helvetica (preferred) or Arial.
- We do not recommend using Adobe Photoshop for designing figures, but we can accept Photoshop generated (.PSD or .TIFF) files only if each element included in the figure (text, labels, pictures, graphs, arrows and scale bars) are on separate layers. All text should be editable in 'type layers' and line-art such as graphs and other simple schematics should be preserved and embedded within 'vector smart objects' - not flattened raster/bitmap graphics.
- Some programs can generate Postscript by 'printing to file' (found in the Print dialogue). If using an application not listed above, save the file in PostScript format or email our Art Editor, Allen Beattie for advice (a.beattie@nature.com).

All placed images (i.e. a photo incorporated into a figure) should be on a separate layer and independent from any superimposed scale bars or text. Individual photographic images must be a minimum of 300+ DPI (at actual size) or kept constant from the original picture acquisition and not

decreased in resolution post image acquisition. All colour artwork should be RGB format.

The total number of Supplementary Figures (not including the “unprocessed scans” Supplementary Figure) should not exceed the number of main display items (figures and/or tables (see our Guide to Authors and March 2012 editorial <http://www.nature.com/ncb/authors/submit/index.html#suppinfo>; <http://www.nature.com/ncb/journal/v14/n3/index.html#ed>). No restrictions apply to Supplementary Tables or Videos, but we advise authors to be selective in including supplemental data.

GUIDELINES FOR EXPERIMENTAL AND STATISTICAL REPORTING

REPORTING REQUIREMENTS – To improve the quality of methods and statistics reporting in our papers we have recently revised the reporting checklist we introduced in 2013. We are now asking all life sciences authors to complete two items: an Editorial Policy Checklist (found here <https://www.nature.com/authors/policies/Policy.pdf>) that verifies compliance with all required editorial policies and a reporting summary (found here <https://www.nature.com/authors/policies/ReportingSummary.pdf>) that collects information on experimental design and reagents. These documents are available to referees to aid the evaluation of the manuscript. Please note that these forms are dynamic 'smart pdfs' and must therefore be downloaded and completed in Adobe Reader. We will then flatten them for ease of use by the reviewers. If you would like to reference the guidance text as you complete the template, please access these flattened versions at <http://www.nature.com/authors/policies/availability.html>.

Author Rebuttal to Initial comments

Reviewer #1

Remarks to the Author:

In this manuscript the authors demonstrate that aSYN phase separation is promoted by an interaction with a positively-charged region near the juxtamembrane region of VAMP2 and dependent on aSYN's capacity to bind to lipid membranes. Notably, co-expression of VAMP2 and aSYN induces the formation of dynamic aSYN-containing condensates in cells. The authors also demonstrate that aSYN condensates accumulate in vesicular structures in cells. Disruption of aSYN lipid binding by the pathological mutation A30P also disrupts condensate formation indicating a possible link between aSYN condensation, SNARE function and disease. The conclusions of the paper are original. It is well-written and conclusions are supported by the presented data, using a range of approaches in vitro and in cells, all of which appear to be rigorously performed with appropriate statistical analysis.

We make the following suggestions:

Point 1 - Please define aSYN 96AAA at first usage.

We have refined this paragraph specifying the precise mutation and citing relevant references from previous literature. Kindly refer to page 7, highlighted in red, for further details.

Point 2 - More useful information should be extracted from the NMR data. For example, the aSYN spectra (Fig 6) demonstrate that the protein is disordered. The clustering of the shifts and the direction suggest a potential stabilization of a helix. Were the intensities of the peaks compared for the datasets? Chemical shifts sizes are not necessarily correlated with interaction strength. Do the intensity changes upon binding reveal any additional interactions between the peptide and the aSYN? In addition, the NMR experiment "1H-15N-BEST-TROSY" should be referenced.

We provided a more detailed discussion of the NMR data and conducted analysis of peak intensities. We have incorporated a new graph depicting relative peak intensities for α -synuclein (aSYN) in the presence of peptide, now presented as Figure 6f and Figure S8c. Please see the details below and refer to page 10, highlighted in red, for the respective changes.

Analysing the alterations in peak intensities, we observed a reduction in peak intensities for C-terminal residues. In addition, we find increased peak intensities at residues E13, G14, V15, and Q99, E104 (relative peak intensity > 1.5), plus decreased peak intensities at residues G7, E20, F94 (relative peak intensity < 0.6). These changes are localized to an early N-terminal region, a second region at the end of the NAC region, and to E104, just preceding the observed decrease in relative peak intensities starting from E105 onwards. These additional changes support the idea that aSYN undergoes a conformational change upon peptide binding, aligning with previous reports demonstrating intramolecular long-range interactions within aSYN (Dedmon et al. 2005; Bertocini et al. 2005; Fernández et al. 2004; Stephens, Zacharopoulou, and Kaminski Schierle 2019) and structural simulation of aSYN reported recently (Parra-Rivas et al. 2023). Furthermore, this also correlates with a recent publication highlighting the occurrence of elongated conformational states during aSYN phase separation (Ubbiali et al. 2022). The presence of multiple changes at the N-terminal and NAC region, both in terms of chemical shift perturbations and relative peak intensities, reflects a high intrinsic disorder with the occurrence of multiple aSYN conformations.

The references for the ^1H - ^{15}N -BEST-TROSY experiment were added to the results section on page 10 and the methods section on page 31 respectively:

- Favier, A. & Brutscher, B. Recovering lost magnetization: Polarization enhancement in biomolecular NMR. *J. Biomol. NMR* 49, 9–15 (2011).
- Solyom, Z. et al. BEST-TROSY experiments for time-efficient sequential resonance assignment of large disordered proteins. *J. Biomol. NMR* 55, 311–321 (2013).

Point 3 - “However, when aSYN YFP was co-expressed with VAMP2, cytosolic aSYN clusters were observed in about 5-10% of the transfected cells” – Is this because the two proteins are only over-expressed in 10% of the cells? Can the authors determine if VAMP2 is not being expressed in cells that don't have the puncta? More information is required on this key point.

Immunocytochemistry (ICC) was conducted to assess the number of cells co-expressing both aSYN YFP and VAMP2. The analysis revealed that 97 to 100% of aSYN YFP-positive cells also express VAMP2. Therefore, although both proteins are present, only a subset of cells exhibit formation of aSYN condensates. For the quantitative analysis of condensate formation, we deliberately have chosen to use systematic image acquisition on the IncuCyte platform rather than individual imaging to mitigate potential bias. Therefore, a certain underestimation of the percentage of cells forming condensates might be the case, nevertheless, it is clearly evident in high-magnification images that only a subset of cells forms condensates. This is likely to be a biological phenomenon, not all of the cells co-expressing aSYN and VAMP2 will develop aSYN condensates. The ICC data have been incorporated as Figure S1, and the corresponding paragraph has been updated accordingly. For detailed information, please refer to page 6, highlighted in red.

Point 4 - Although 1,6-hexanediol is often used to disrupt condensates, it is not a general condensate dissolver. It seems an unlikely candidate for disrupting electrostatic interactions. It suggests that electrostatic, polar/H-bonding and hydrophobic interactions contribute to condensate formation in this case. Alternatively, this 1,6-hexanediol effect might be non-specific.

Thank you for highlighting this important point. We have now provided clarification in the text, explicitly stating that 1,6-hexanediol disrupts hydrophobic interactions. It's crucial to note that 1,6-hexanediol is unlikely to interfere with the electrostatic interactions (between alpha-synuclein and VAMP2), however it will interfere with hydrophobic interactions, particularly contributed via alpha-synuclein's NAC region.

To further support the notion that the observed puncta signify condensates, we added new data in addition to the 1,6-hexanediol experiments and the FRAP experiments, which show fusion and fission events in time-resolved experiments, revealing fluid-like behaviour of the observed condensates. We have updated Figure 2 to include images depicting droplet fusion and fission (Figure 2e), and corresponding movies have been added as Supplementary Material. The paragraph addressing these findings has been revised; please refer to page 6/7, highlighted in red, for the updated information.

Point 5 - “To test the relevance of lipid binding we used aSYN A30P, an aSYN disease variant 79 with defective lipid binding 31,38,80–82 little effect of A30P.” – This sentence is unclear.

The relevant sentence has been amended; please refer to page 11 for the highlighted correction.

Point 6 - “This is interesting since immunoprecipitation experiments show that the N-terminal proline-rich domain of VAMP2 mediates VAMP2/aSYN interaction” -- The data do not indicate that the N-terminal proline rich region does not contribute to phase separation within the context of the full protein, only that the relatively small peptides used do not promote phase separation. With respect to figure 5, it should be noted that short peptides, which are important for phase separation in the context of the protein, can actually impede phase separation when extracted from the larger protein by acting as “caps”. If they have a valence of 1, they can block interactions that contribute to multivalency (see Sanders, D., Kedersha, N. et al. Cell 2020, 181:2 for an explanation). This is another reason to analyze the intensity change in the NMR spectra as they will provide information on interaction not visible in the chemical shift perturbations. For disordered proteins, chemical shift perturbations often rely on large chemical shift changes effected by changes in secondary structure as they are typically in the fast exchange regime. In fact, for many disordered protein interactions, no chemical shift changes are observed, though intensity changes can be dramatic.

In the text, we have provided clarification that our findings based on the N-terminal peptides are specifically confined to the experiment illustrated in Figure 5. We agree that it is important to note that these results do not preclude the potential contribution of the N-terminal proline-rich region to the phase separation of the full-length protein. Additionally, we elaborate on the potential role of the peptide acting as a cap therewith omitting further network interactions. Please refer to page 14/15, highlighted in red, for the respective changes. For insights into the NMR data analysis, kindly review Point 2.

Point 7 - “LLPS” should not be used as a keyword since it is not used in the manuscript and the data are not definitive that this is simple LLPS. Many biomolecular condensates use more complex phase separation processes (see DOI: 10.1016/j.molcel.2022.05.018), and the manuscript, appropriately, refers only to “phase separation”. Thus, the keyword should be “phase separation”.

“LLPS” was removed as a keyword. We agree, that aSYN is undergoing a more complex phase separation process. “Phase separation” is included as keyword.

Reviewer #2**Remarks to the Author:**

The authors co-express alpha-synuclein (aSyn) with a panel of Parkinson's Disease associated proteins in HeLa cells and find that one of them, VAMP2, leads to the formation of phase-separated condensates. In vitro, VAMP2 promotes liquid-liquid phase separation (LLPS) of aSyn, with which it forms co-condensates. A 14-residue peptide representing the juxtamembrane region of VAMP2 strongly enhanced aSyn LLPS, even in the absence of non-physiologically high levels of calcium which was otherwise required. NMR experiments demonstrate that this VAMP2 peptide primarily interacts with the C-terminal portion of aSyn. In vivo, LLPS appears to depend on the membrane association of aSyn, as it is abolished by an aSyn mutant (A30P) with reduced membrane binding. EM reveals cytosolic accumulation of heterogeneously sized vesicles in cells expressing both aSyn and VAMP2, consistent with the idea that aSyn and VAMP2 form membrane associated condensates.

Point 1 - These results are interesting and they do, in my judgment, advance the field. My main concern is whether we learn enough about the relationship between aSyn:VAMP2 LLPS and synaptic vesicle cycling to justify publication in Nature Cell Biology. As suggested by the authors, the link between aSyn phase separation and VAMP2 is intriguing and hints at a possible regulatory mechanism linked to SNARE assembly/disassembly. It will be interesting to test this hypothesis by taking advantage of the structural insights gained in this work.

We conducted new experiments to investigate whether the formation of α -synuclein (aSYN) condensates has an impact on the assembly of the SNARE complex. While we find a trend towards increased SNARE complex formation, where 4 out of 5 repeats show higher SNARE complexes formation upon wild-type aSYN co-expression, we do not observe a difference with the respective aSYN variants with decreased/abolished condensate formation (aSYN 96AAA and A30P aSYN). This suggests that aSYN is able to affect SNARE complex assembly but an additional effect upon condensate formation in our model system is too small. The data for all repeats of the SNARE complex assembly experiment, including all unprocessed images of the blots are attached to be reviewed (Figure 1 below).

A functional role of aSYN phase separation is reinforced by new data we obtained through Correlative light and Electron Microscopy (CLEM). By combining fluorescence imaging of condensates with ultrastructural visualization by electron microscopy, we demonstrate that aSYN condensates contain about 10 and 50 vesicles each. Notably, this vesicle clustering by aSYN is distinct from the vesicle clustering observed in synapsin-1 condensates (Park et al. 2021), suggesting a complementary role for aSYN. The CLEM imaging highlights the role of aSYN in vesicle clustering and for the first time demonstrates that aSYN condensates

maintain vesicles not only in vitro (Hardenberg et al. 2021) but that this function extends to its role within a biological context. The respective CLEM images and analysis have been added as Figures 7g/h/i and we discuss the role of aSYN on different vesicle entities further in the text.

Despite these insights, the initiation of aSYN phase separation by VAMP2 hints at a potential role during vesicle exocytosis. In further exploration, we reveal that aSYN condensates specifically attract complexin-1 and complexin-2, synaptic proteins mediating vesicle release. In particular, we find that complexin-2, predominantly expressed in excitatory neurons, is more enriched compared to complexin-1, mainly expressed in inhibitory neurons (S. L. Eastwood, Cotter, and Harrison 2001; Sharon L. Eastwood and Harrison 2001; Harrison and Eastwood 1998). This distribution corresponds with a predominant expression of aSYN in excitatory neurons (Taguchi et al. 2019). These findings are especially interesting since complexin upregulation has been observed in aSYN knock-out and mutant mice (Chandra et al. 2004; Gispert et al. 2015). Furthermore, complexin has been linked to human disease, such as Parkinson's disease and other synucleinopathies (Lahut et al. 2017; Nilsson et al. 2023). Please refer to page 12, highlighted in red, for details on CLEM and complexin.

We trust that these revisions significantly contribute to the overall strength and coherence of our manuscript. The function of aSYN has remained an open question since its initial association with Parkinson's disease in 1997 (Spillantini et al. 1997). Our manuscript, presenting the molecular mechanism regulating aSYN phase separation in its biological context, demonstrates a critical advancement in the field and opens new ways to study and understand its intricate function. We provide new evidence that aSYN phase separation regulates vesicles and co-condensation of complexin-1 and 2, which may evolve as important players for the normal function of aSYN and its dysfunction in synucleinopathies. We, therefore, anticipate that future research will build upon the data we present here. We look forward to the opportunity to further discuss our work and herewith resubmit the revised manuscript for your review. Thank you for your time and consideration.

Sincerely,

Dr. Janin Lautenschläger

Principal Investigator & Royal Society Dorothy Hodgkin Research Fellow

University of Cambridge

Relevant References:

- Bertoncini, Carlos W., Young Sang Jung, Claudio O. Fernandez, Wolfgang Hoyer, Christian Griesinger, Thomas M. Jovin, and Markus Zweckstetter. 2005. "Release of Long-Range Tertiary Interactions Potentiates Aggregation of Natively Unstructured α -Synuclein." *Proceedings of the National Academy of Sciences of the United States of America* 102 (5): 1430–35. <https://doi.org/10.1073/pnas.0407146102>.
- Chandra, Sreenganga, Francesco Fornai, Hyung Bae Kwon, Umar Yazdani, Deniz Atasoy, Xinran Liu, Robert E. Hammer, et al. 2004. "Double-Knockout Mice for α - and β -Synucleins: Effect on Synaptic Functions." *Proceedings of the National Academy of Sciences of the United States of America* 101 (41): 14966–71. <https://doi.org/10.1073/pnas.0406283101>.
- Dedmon, Matthew M., Kresten Lindorff-Larsen, John Christodoulou, Michele Vendruscolo, and Christopher M. Dobson. 2005. "Mapping Long-Range Interactions in α -Synuclein Using Spin-Label NMR and Ensemble Molecular Dynamics Simulations." *Journal of the American Chemical Society* 127 (2): 476–77. <https://doi.org/10.1021/ja044834j>.
- Eastwood, S. L., D. Cotter, and P. J. Harrison. 2001. "Cerebellar Synaptic Protein Expression in Schizophrenia." *Neuroscience* 105 (1): 219–29. [https://doi.org/10.1016/S0306-4522\(01\)00141-5](https://doi.org/10.1016/S0306-4522(01)00141-5).
- Eastwood, Sharon L., and Paul J. Harrison. 2001. "Synaptic Pathology in the Anterior Cingulate Cortex in Schizophrenia and Mood Disorders. A Review and a Western Blot Study of Synaptophysin, GAP-43 and the Complexins." *Brain Research Bulletin* 55 (5): 569–78. [https://doi.org/10.1016/S0361-9230\(01\)00530-5](https://doi.org/10.1016/S0361-9230(01)00530-5).
- Fernández, Claudio O., Wolfgang Hoyer, Maricus Zweckstetter, Elizabeth A. Jares-Erijman, Vinod Subramaniam, Christian Griesinger, and Thomas M. Jovin. 2004. "NMR of α -Synuclein-Polyamine Complexes Elucidates the Mechanism and Kinetics of Induced Aggregation." *EMBO Journal* 23 (10): 2039–46. <https://doi.org/10.1038/sj.emboj.7600211>.
- Gispert, Suzana, Alexander Kurz, Nadine Brehm, Katrin Rau, Michael Walter, Olaf Riess, and Georg Auburger. 2015. "Complexin-1 and Foxp1 Expression Changes Are Novel Brain Effects of Alpha-Synuclein Pathology." *Molecular Neurobiology* 52 (1): 57–63. <https://doi.org/10.1007/s12035-014-8844-0>.
- Hardenberg, Maarten C, Tessa Sinnige, Sam Casford, Samuel Dada, Chetan Poudel, Elizabeth A Robinson, Monika Fuxreiter, et al. 2021. "Observation of an α -Synuclein Liquid Droplet State and Its Maturation into Lewy Body-like Assemblies." *Journal of Molecular Cell Biology* 13: 282–94. <https://doi.org/10.1093/jmcb/mjaa075>.
- Harrison, Paul J., and Sharon L. Eastwood. 1998. "Preferential Involvement of Excitatory Neurons in Medial Temporal Lobe in Schizophrenia." *Lancet* 352 (9141): 1669–73. [https://doi.org/10.1016/S0140-6736\(98\)03341-8](https://doi.org/10.1016/S0140-6736(98)03341-8).
- Lahut, Suna, Suzana Gispert, Özgür Ömür, Candan Depboylu, Kay Seidel, Jorge Antolio Dominguez Bautista, Nadine Brehm, et al. 2017. "Blood RNA Biomarkers in Prodromal PARK4 and Rapid Eye Movement Sleep Behavior Disorder Show Role of Complexin 1 Loss for Risk of Parkinson's Disease." *DMM Disease Models and Mechanisms* 10 (5): 619–31. <https://doi.org/10.1242/dmm.028035>.
- Nilsson, Johanna, Julius Constantinescu, Bengt Nellgård, Protik Jakobsson, Wagner S. Brum, Johan Gobom, Lars Forsgren, et al. 2023. "Cerebrospinal Fluid Biomarkers of Synaptic Dysfunction Are Altered in Parkinson's

- Disease and Related Disorders." *Movement Disorders* 38 (2): 267–77. <https://doi.org/10.1002/mds.29287>.
- Park, Daehun, Yumei Wu, Sang Eun Lee, Goeun Kim, Seonyoung Jeong, Dragomir Milovanovic, Pietro De Camilli, and Sunghoe Chang. 2021. "Cooperative Function of Synaptophysin and Synapsin in the Generation of Synaptic Vesicle-like Clusters in Non-Neuronal Cells." *Nature Communications* 12 (1). <https://doi.org/10.1038/s41467-020-20462-z>.
- Parra-Rivas, Leonardo A, Kayalvizhi Madhivanan, Lina Wang, Nicholas P Boyer, Dube Dheeraj, Brent D Aulston, Donald P Pizzo, et al. 2023. "Serine-129 Phosphorylation of α -Synuclein Is a Trigger for Physiologic Protein-Protein Interactions and Synaptic Function." *Neuron* 111 (24): 1–22. <https://doi.org/10.1016/j.neuron.2023.11.020>.
- Spillantini, Maria Grazia, Marie Luise Schmidt, Virginia M.-Y. Lee, John Q. Trojanowski, Ross Jakes, and Michel Goedert. 1997. "A-Synuclein in Lewy Bodies." *Nature* 388: 839–40.
- Stephens, Amberley D., Maria Zacharopoulou, and Gabriele S. Kaminski Schierle. 2019. "The Cellular Environment Affects Monomeric α -Synuclein Structure." *Trends in Biochemical Sciences* 44 (5): 453–66. <https://doi.org/10.1016/j.tibs.2018.11.005>.
- Taguchi, Katsutoshi, Yoshihisa Watanabe, Atsushi Tsujimura, and Masaki Tanaka. 2019. "Expression of α -Synuclein Is Regulated in a Neuronal Cell Type-Dependent Manner." *Anatomical Science International* 94 (1): 11–22. <https://doi.org/10.1007/s12565-018-0464-8>.
- Ubbiali, Daniele, Marta Fratini, Lolita Piersimoni, Christian Ihling, Marc Kipping, Ingo Heilmann, Claudio Iacobucci, and Andrea Sinz. 2022. "Direct Observation of 'Elongated' Conformational States in A-Synuclein upon Liquid-Liquid Phase Separation." *Angewandte Chemie* 202205726: 1–6. <https://doi.org/10.1002/ange.202205726>.

Figure 1

Figure 1. Evaluation of SNARE complex assembly upon co-expression of Syntaxin-1, SNAP25, VAMP2 and aSYN

- SNARE complex formation shown as Syntaxin-1 positive bands with a molecular weight above 70kDa. Bands are absent in the un-transfected control and in the boiled samples (Fig. 1d). The same membrane shows Syntaxin-1 monomer and aSYN band.
- Quantification of SNARE complexes relative to Syntaxin-1 shows a trend towards increased SNARE complex assembly for WT aSYN compared to mScarlet. No difference for the 96AAA aSYN variant with decreased condensate formation and the A30P aSYN variant with no condensate formation is seen.
- Quantification of SNARE complexes relative to aSYN. No consistent difference for the 96AAA aSYN variant with decreased condensate formation and the A30P aSYN variant with no condensate formation is seen.

- d) Bands showing reduction of SNARE complexes after boiling and the monomeric bands for all 3 SNARE proteins after boiling to confirm co-expression.
- e) Respective unprocessed PVDF membranes, 5 biological repeats.

e Repeat 1

Repeat 2

Repeat 3

Repeat4

Repeat 5

SNARE complex assembly

HeLa cells were plated at 50,000 cells/well in 12-well plates (353043, Falcon) and transfected the following day using Fugene HD Transfection reagent (E2311, Promega) as described above. Briefly, per reaction 125 μ L OptiMEM (31985-062, Gibco) were set up in 1.5 mL sterile Eppendorf tubes. 125 ng of DNA per plasmid and 2.5 μ L of Fugene reagent were added. VAMP2, Syntaxin, SNAP25 and the respective aSYN variant, or mScarlet control were used for each reaction. The transfection mix was incubated for 15 min at room temperature, added onto the cells for 1 min and then topped up with 1000 μ L complete media. The following day cells were washed 3 times with PBS and lysed in 60 μ L 1x PBS with 0.1% TritonX-100 (Fisher Scientific) supplemented with protease and phosphatase inhibitors (A32955 and A32957, Thermo Scientific). Protein concentrations were measured using BCA assay (AR0146-A/B, Boster Bio Technology Co LTD, Pleasanton, USA). 20 μ g of lysate, unboiled, or after boiling for 20 min at 95 °C to denature SNARE complexes, were loaded onto 4-12% Bis-Tris SDS-PAGE gels (mPAGE, MP41G12, Millipore, Darmstadt, Germany) and blotted on PVDF membrane (IB24001, Invitrogen) using the iBlot2 system (23V, 6 min, Invitrogen, Massachusetts, USA). Membranes were subsequently incubated with 4% paraformaldehyde / 0.1% glutaraldehyde in PBS for 30 min at room temperature, blocked in 5% BSA 1xTBS 0.1% Tween20, and incubated with primary antibody at 4 °C overnight. SNARE complexes were detected using a primary antibody specific to Syntaxin (110111, Synaptic Systems, Göttingen, Germany). Levels of aSYN were detected via a pan-aSYN antibody (ab6167, Abcam) or mCherry antibody (A85306, antibodies.com, Cambridge, UK) to serve as loading controls. Boiled samples were stained using primary antibodies against Syntaxin (110111, Synaptic Systems), SNAP25 (111002, Synaptic Systems) and Flag (20543-1-AP, Proteintech) to

detect VAMP2-FLAG. All primary antibodies were added at 1:1000 in blocking solution. After four washes in 1xTBS 0.1% Tween20, secondary antibodies were stained for 1h at room temperature. Goat anti-mouse DyLight™ 800 and goat anti-rabbit DyLight™ 680 (SA535521 and 35568, Invitrogen) were applied, all used at 1:10000 in blocking solution. After another four washes in 1xTBS 0.1% Tween20 the membranes were imaged on Odyssey CLx (LI-COR, Lincoln, USA). Analysis of the respective bands was performed in FIJI.

Decision Letter, first revision:

Our ref: NCB-A51699A

22nd March 2024

Dear Dr. Lautenschläger,

Thank you for submitting your revised manuscript "VAMP2 regulates phase separation of alpha-synuclein" (NCB-A51699A). It has now been seen by the original referees and their comments are below. The reviewers find that the paper has improved in revision, and therefore we'll be happy in principle to publish it in Nature Cell Biology, pending textual revisions to satisfy the referees' final requests and to comply with our editorial and formatting guidelines.

Thank you again for your interest in Nature Cell Biology Please do not hesitate to contact me if you have any questions.

Sincerely,
Daryl

Daryl Jason Verzosa David, PhD

Senior Editor, Nature Cell Biology
Nature Portfolio
Advisory Editor, npj Biological Physics and Mechanics

Heidelberger Platz 3, 14197 Berlin, Germany
Email: daryl.david@nature.com
ORCID: <https://orcid.org/0000-0002-9253-4805>

Reviewer #1 (Remarks to the Author):

The revised manuscript incorporated many of our suggestions and also added some interesting CLEM (Correlative Light and Electron Microscopy) data which is interesting. The work is a valuable contribution to the field.

A minor point is that while 1,6-hexanediol may be described in the literature as competing with hydrophobic interactions, its effect is likely more complex. Changes in solvent nature are likely and specific inhibition of enzymes by the compound have also been reported, with other related alcohols not able to mimic these effects. This reviewer strongly suggest skipping the interpretation and just reporting the results.

Reviewer #2 (Remarks to the Author):

The revised manuscript is improved in some ways, as noted by the authors in their points i-iii at the top of their rebuttal. And, as I wrote in my original review, I do think this work moves the field forward. Its implications for synaptic vesicle biology are, however, still unclear. More broadly, the authors point out in the introduction that "the physiological relevance of aSYN [α -synuclein] phase separation has not been demonstrated" and, in my view, that's still true. Whether to move forward with this manuscript is therefore, and inevitably, an editorial decision.

Regardless, having now spent some hours reading and rereading the work, I will offer a few suggestions, none of which entail additional experiments.

1. The abstract proposes "a potential switch from the dispersed to the phase-separated state during vesicle recycling". This seems entirely speculative – can the authors propose a mechanism?
2. I'm pretty sure the last word of the abstract should be "partners" (plural).
3. On p. 3, the authors state that "aSYN localizes to synapsin condensates in cells (ref. 27)". Do they still think so?
4. Has anyone shown that aSYN phase separates in axon terminals?
5. A really surprising finding is that aSYN phase separation with and without VAMP 1-96 is almost indistinguishable (Fig. 4b). (And, incidentally, the measurement of Csat seems implausibly precise.) How do the authors think about this?

6. On pp. 8-9, the authors refer multiple times to peptides “resembling” residues X-Y of VAMP2. Why resembling? Do they just mean that the peptides have N- and C-termini, which the residues do not? Or are there sequence differences (and if so, why)?
7. Another surprising finding is that the scrambled “JMD long” peptide shows relatively robust activity in promoting aSYN phase separation (Fig. 5f). Am I thinking about the data incorrectly? Could this be consistent with a model in which the dominant effect is simply the number of positive charges (noting that spermine also promotes phase separation)?
8. On p. 11, the authors refer to “a secondary uptake of vesicles.” What is this?
9. On p. 12, the authors write, “Therefore, our findings indicating a complementary function for aSYN condensates in mediating vesicle clustering.” First, complementary to what? Second, what sort of vesicles?
10. Also on p. 12, the authors write, “Together this underlines the biological relevance of aSYN condensate formation....” Could they be more explicit?
11. Continuing, “...and suggests that the liquid phase-separated state mediates synaptic function via regulation of vesicles and interaction partners, such as complexins.” Again, could they be more explicit?
12. On p. 14, the authors write: “Further to that, we show that aSYN condensates in cells have a regulatory role in clustering vesicles.” What is the regulatory role, and how is it demonstrated in this manuscript?

Decision Letter, final checks:

Our ref: NCB-A51699A

10th April 2024

Dear Dr. Lautenschläger,

Thank you for your patience as we’ve prepared the guidelines for final submission of your Nature Cell Biology manuscript, “VAMP2 regulates phase separation of alpha-synuclein” (NCB-A51699A). Please carefully follow the step-by-step instructions provided in the attached file, and add a response in each row of the table to indicate the changes that you have made. Ensuring that each point is addressed will help to ensure that your revised manuscript can be swiftly handed over to our production team.

In recognition of the time and expertise our reviewers provide to Nature Cell Biology's editorial process, we would like to formally acknowledge their contribution to the external peer review of your manuscript entitled "VAMP2 regulates phase separation of alpha-synuclein". For those reviewers who give their assent, we will be publishing their names alongside the published article.

Nature Cell Biology offers a Transparent Peer Review option for new original research manuscripts submitted after December 1st, 2019. As part of this initiative, we encourage our authors to support increased transparency into the peer review process by agreeing to have the reviewer comments, author rebuttal letters, and editorial decision letters published as a Supplementary item. When you submit your final files please clearly state in your cover letter whether or not you would like to participate in this initiative. Please note that failure to state your preference will result in delays in accepting your manuscript for publication.

Cover suggestions

COVER ARTWORK: We welcome submissions of artwork for consideration for our cover. For more information, please see our guide for cover artwork.

Nature Cell Biology has now transitioned to a unified Rights Collection system which will allow our Author Services team to quickly and easily collect the rights and permissions required to publish your work. Approximately 10 days after your paper is formally accepted, you will receive an email in providing you with a link to complete the grant of rights. If your paper is eligible for Open Access, our Author Services team will also be in touch regarding any additional information that may be required to arrange payment for your article.

Please note that *Nature Cell Biology* is a Transformative Journal (TJ). Authors may publish their research with us through the traditional subscription access route or make their paper immediately open access through payment of an article-processing charge (APC). Authors will not be required to make a final decision about access to their article until it has been accepted. Find out more about Transformative Journals

Please use the following link for uploading these materials:
[Redacted]

Best regards,

Kendra Donahue
Staff
Nature Cell Biology

On behalf of

Daryl Jason Verzosa David, PhD

Senior Editor, Nature Cell Biology
Nature Portfolio
Advisory Editor, npj Biological Physics and Mechanics

Heidelberger Platz 3, 14197 Berlin, Germany
Email: daryl.david@nature.com
ORCID: <https://orcid.org/0000-0002-9253-4805>

Reviewer #1:

Remarks to the Author:

The revised manuscript incorporated many of our suggestions and also added some interesting CLEM (Correlative Light and Electron Microscopy) data which is interesting. The work is a valuable contribution to the field.

A minor point is that while 1,6-hexanediol may be described in the literature as competing with hydrophobic interactions, its effect is likely more complex. Changes in solvent nature are likely and specific inhibition of enzymes by the compound have also been reported, with other related alcohols not able to mimic these effects. This reviewer strongly suggest skipping the interpretation and just reporting the results.

Reviewer #2:

Remarks to the Author:

The revised manuscript is improved in some ways, as noted by the authors in their points i-iii at the top of their rebuttal. And, as I wrote in my original review, I do think this work moves the field forward. Its implications for synaptic vesicle biology are, however, still unclear. More broadly, the authors point out in the introduction that "the physiological relevance of aSYN [α -synuclein] phase separation has not been demonstrated" and, in my view, that's still true. Whether to move forward with this manuscript is therefore, and inevitably, an editorial decision.

Regardless, having now spent some hours reading and rereading the work, I will offer a few suggestions, none of which entail additional experiments.

1. The abstract proposes "a potential switch from the dispersed to the phase-separated state during vesicle recycling". This seems entirely speculative – can the authors propose a mechanism?
2. I'm pretty sure the last word of the abstract should be "partners" (plural).
3. On p. 3, the authors state that "aSYN localizes to synapsin condensates in cells (ref. 27)". Do they still think so?
4. Has anyone shown that aSYN phase separates in axon terminals?
5. A really surprising finding is that aSYN phase separation with and without VAMP 1-96 is almost indistinguishable (Fig. 4b). (And, incidentally, the measurement of Csat seems implausibly precise.) How do the authors think about this?
6. On pp. 8-9, the authors refer multiple times to peptides "resembling" residues X-Y of VAMP2. Why resembling? Do they just mean that the peptides have N- and C-termini, which the residues do not? Or are there sequence differences (and if so, why)?
7. Another surprising finding is that the scrambled "JMD long" peptide shows relatively robust activity in promoting aSYN phase separation (Fig. 5f). Am I thinking about the data incorrectly? Could this be consistent with a model in which the dominant effect is simply the number of positive charges (noting that spermine also promotes phase separation)?
8. On p. 11, the authors refer to "a secondary uptake of vesicles." What is this?
9. On p. 12, the authors write, "Therefore, our findings indicating a complementary function for aSYN condensates in mediating vesicle clustering." First, complementary to what? Second, what sort of vesicles?
10. Also on p. 12, the authors write, "Together this underlines the biological relevance of aSYN condensate formation...." Could they be more explicit?
11. Continuing, "...and suggests that the liquid phase-separated state mediates synaptic function via regulation of vesicles and interaction partners, such as complexins." Again, could they be more explicit?

12. On p. 14, the authors write: "Further to that, we show that aSYN condensates in cells have a regulatory role in clustering vesicles." What is the regulatory role, and how is it demonstrated in this manuscript?

Author Rebuttal, first revision:

Reviewer #1 (Remarks to the Author)

The revised manuscript incorporated many of our suggestions and also added some interesting CLEM (Correlative Light and Electron Microscopy) data which is interesting. The work is a valuable contribution to the field.

A minor point is that while 1,6-hexanediol may be described in the literature as competing with hydrophobic interactions, its effect is likely more complex. Changes in solvent nature are likely and specific inhibition of enzymes by the compound have also been reported, with other related alcohols not able to mimic these effects. This reviewer strongly suggest skipping the interpretation and just reporting the results.

We corrected the respective paragraph on page 6/7, all sections discussing the hydrophobic effect of 1,6-hexanediol have been removed. We now only refer to the reversibility of aSYN clusters.

Reviewer #2 (Remarks to the Author)

The revised manuscript is improved in some ways, as noted by the authors in their points i-iii at the top of their rebuttal. And, as I wrote in my original review, I do think this work moves the field forward. Its implications for synaptic vesicle biology are, however, still unclear. More broadly, the authors point out in the introduction that "the physiological relevance of aSYN [α -synuclein] phase separation has not been demonstrated" and, in my view, that's still true. Whether to move forward with this manuscript is therefore, and inevitably, an editorial decision.

Regardless, having now spent some hours reading and rereading the work, I will offer a few suggestions, none of which entail additional experiments.

1. The abstract proposes “a potential switch from the dispersed to the phase-separated state during vesicle recycling”. This seems entirely speculative – can the authors propose a mechanism?

The abstract has been revised upon editorial suggestion. With this the sentence “a potential switch from the dispersed to the phase-separated state during vesicle recycling” has been removed.

2. I’m pretty sure the last word of the abstract should be “partners” (plural).

The word “partners” was removed from the abstract upon revision.

3. On p. 3, the authors state that “aSYN localizes to synapsin condensates in cells (ref. 27)”. Do they still think so?

This statement refers to a manuscript which was published by another research group working on synapsin and aSYN. We believe it is essential to acknowledge their work as it is the first to explore aSYN in a phase separation context within cells. Furthermore, recent papers identified a specific interaction of synapsin with aSYN (Parra-Rivas et al. 2023; Stavsky et al. 2023), which supports the idea of an interlink between synapsin and aSYN phase separation.

4. Has anyone shown that aSYN phase separates in axon terminals?

Thus far, phase separation of aSYN has not been shown at the axon terminal. However, it is likely that the observed enrichment of aSYN at the synapse demonstrates phase separation, as mere membrane surface enrichment may not achieve the necessary concentration gradient/ratio difference. Yet, methodologies for assessing phase separation are limited, particularly at the synapse due to its small size. Notably, phase separation at the synapse has been demonstrated for another protein, synapsin-1, through antibody microinjection targeting its IDR region (Pechstein et al. 2020). Investigating aSYN phase separation, delineating the regions and mechanisms involved in its phase separation is essential for conducting analogous studies. The findings presented here establish the groundwork for such investigations.

5. A really surprising finding is that aSYN phase separation with and without VAMP 1-96 is almost indistinguishable (Fig. 4b). (And, incidentally, the measurement of Csat seems implausibly precise.) How do the authors think about this?

In Fig. 4b a notable alteration in aSYN Csat is evident, decreasing from 36.6 +/- 4.7 μ M without VAMP96 to 26.4 +/- 1.6 μ M in the presence of 10 μ M VAMP96. This effect appears modest in the gel, however our assay necessitated working with low concentrations of VAMP96, as VAMP96 also demonstrates a tendency to form droplets.

6. On pp. 8-9, the authors refer multiple times to peptides “resembling” residues X-Y of VAMP2. Why resembling? Do they just mean that the peptides have N- and C-termini, which the residues do not? Or are there sequence differences (and if so, why)?

The term “resembling” has been removed and substituted with "peptides identical to amino residues". The peptides precisely matched the sequence of the specified protein segments. For respective changes in the text, please refer to pages 9 and 14.

7. Another surprising finding is that the scrambled “JMD long” peptide shows relatively robust activity in promoting aSYN phase separation (Fig. 5f). Am I thinking about the data incorrectly? Could this be consistent with a model in which the dominant effect is simply the number of positive charges (noting that spermine also promotes phase separation)?

Indeed, there is an electrostatic effect facilitated by positive charges on aSYN phase separation. However, what we show with this experiment is that the same amount of charges, when scrambled in a random order, did not produce the same effect as the VAMP2 JMD identical peptide. It is not anticipated that the scrambled peptide has no effect on aSYN phase separation, as it will contribute to its phase separation via positive charges. However, with a smaller effect seen, this demonstrates that aSYN phase separation is not solely mediated by the number of charges. Please see page 9 for respective changes in the text.

8. On p. 11, the authors refer to “a secondary uptake of vesicles.” What is this?

We use the term "secondary uptake of vesicles" to describe that the aSYN liquid phase seems not to assemble on the surface of synaptotagmin vesicles. This is evidenced by the lack of colocalization between the brightest synaptotagmin punctae and aSYN condensates. Instead, we find that aSYN condensates colocalise with lower intensity synaptotagmin punctae, suggesting that they attract vesicles over time. Our terminology in the text has been revised accordingly. Please refer to page 11 for the updated paragraph.

9. On p. 12, the authors write, "Therefore, our findings indicating a complementary function for aSYN condensates in mediating vesicle clustering." First, complementary to what? Second, what sort of vesicles?

We use the term "complementary function" to describe aSYN's role in vesicle clustering, which is distinct from synapsin's function in clustering reserve pool vesicles. We have clarified this in the paragraph on page 12. While synapsin condensates have been demonstrated to cluster vesicles of very uniform size (Park et al. 2021), our findings indicate that aSYN condensates cluster vesicles of various sizes. In our correlative light and electron microscopy (CLEM) experiments, we have identified small vesicles, which could resemble early endosomes, but we also find larger vesicles and autophagolysosomes. Although we attempted to further characterize these vesicle types, we observed colocalization with a variety of vesicle markers, possibly due to limitations of our co-expression system with VAMP2. However, identifying the vesicle type may provide insights into the functional significance of aSYN phase separation. Nonetheless, it's worth noting that the vesicle clustering mediated by aSYN condensates may inherently be less specific, involving vesicles of various sizes.

10. Also on p. 12, the authors write, "Together this underlines the biological relevance of aSYN condensate formation...." Could they be more explicit?

11. Continuing, "...and suggests that the liquid phase-separated state mediates synaptic function via regulation of vesicles and interaction partners, such as complexins." Again, could they be more explicit?

12. On p. 14, the authors write: "Further to that, we show that aSYN condensates in cells have a regulatory role in clustering vesicles." What is the regulatory role, and how is it demonstrated in this manuscript?

The questions above relate to the functional role of aSYN phase separation during synaptic transmission and we acknowledge that understanding the regulatory role and biological significance of aSYN phase separation requires further investigation. Since its initial association with Parkinson's disease in 1997 (Spillantini et al. 1997) the function of aSYN has remained an open question. While our manuscript does

not provide definitive evidence of its exact function, it elucidates the molecular mechanism governing aSYN phase separation within a biological context. This represents a significant step towards unravelling its role at the synapse i.e. via identification of the vesicle species entrapped in aSYN condensates or phenotypes evolving upon modulation of aSYN phase separation.

We rephrased the respective sentences on page 12 and 14. Furthermore, with our discussion we highlight potential working models and routes to explore in the context with other previous and recent findings.

We trust that these revisions significantly contribute to the overall strength and coherence of our manuscript. Please let us know if any additional changes are required. Thank you for your time and consideration.

Sincerely,

Dr. Janin Lautenschläger
Principal Investigator & Royal Society Dorothy Hodgkin Research Fellow
University of Cambridge

References:

Park, Daehun, Yumei Wu, Sang Eun Lee, Goeun Kim, Seonyoung Jeong, Dragomir Milovanovic, Pietro De Camilli, and Sunghoe Chang. 2021. "Cooperative Function of Synaptophysin and Synapsin in the Generation of Synaptic Vesicle-like Clusters in Non-Neuronal Cells." *Nature Communications* 12 (1). <https://doi.org/10.1038/s41467-020-20462-z>.

Parra-Rivas, Leonardo A, Kayalvizhi Madhivanan, Lina Wang, Nicholas P Boyer, Dube Dheeraj, Brent D Aulston, Donald P Pizzo, et al. 2023. "Serine-129 Phosphorylation of α -Synuclein Is a Trigger for Physiologic Protein-Protein Interactions and Synaptic Function." *Neuron* 111 (24): 1–22. <https://doi.org/10.1016/j.neuron.2023.11.020>.

Pechstein, Arndt, Nikolay Tomilin, Kristin Fredrich, Olga Vorontsova, Elena Sopova, Emma Evergren, Volker Haucke, Lennart Brodin, and Oleg Shupliakov. 2020. "Vesicle Clustering in a Living Synapse Depends on a Synapsin Region That Mediates Phase Separation." *Cell Reports* 30 (8): 2594-2602.e3. <https://doi.org/10.1016/j.celrep.2020.01.092>.

Spillantini, Maria Grazia, Marie Luise Schmidt, Virginia M.-Y. Lee, John Q. Trojanowski, Ross Jakes, and Michel Goedert. 1997. "A-Synuclein in Lewy Bodies." *Nature* 388: 839–40.

<https://doi.org/10.1038/42166>.

Stavsky, Alexandra, Leonardo A Parra-rivas, Shani Tal, Kayalvizhi Madhivanan, and Subhojit Roy. 2023. "Synapsin E-Domain Is Essential for α -Synuclein Function." *Elife*, 1–11.

<https://doi.org/10.1101/2023.06.24.546170>.

Final Decision Letter:

Dear Dr Lautenschläger,

I am pleased to inform you that your manuscript, "VAMP2 regulates phase separation of α -synuclein", has now been accepted for publication in *Nature Cell Biology*.

Over the next few weeks, your paper will be copyedited to ensure that it conforms to *Nature Cell Biology* style. Once your paper is typeset, you will receive an email with a link to choose the appropriate publishing options for your paper and our Author Services team will be in touch regarding any additional information that may be required.

Once your paper has been scheduled for online publication, the Nature press office will be in touch to confirm the details. An online order form for reprints of your paper is available at <https://www.nature.com/reprints/author-reprints.html>. All co-authors, authors' institutions and

authors' funding agencies can order reprints using the form appropriate to their geographical region.

Please note that *Nature Cell Biology* is a Transformative Journal (TJ). Authors may publish their research with us through the traditional subscription access route or make their paper immediately open access through payment of an article-processing charge (APC). Authors will not be required to make a final decision about access to their article until it has been accepted. Find out more about Transformative Journals

If you have not already done so, we strongly recommend that you upload the step-by-step protocols used in this manuscript to protocols.io (<https://protocols.io>), an open online resource that allows researchers to share their detailed experimental know-how. All uploaded protocols are made freely available and are assigned DOIs for ease of citation. Protocols and Nature Portfolio journal papers in which they are used can be linked to one another, and this link is clearly and prominently visible in the online versions of both. Authors who performed the specific experiments can act as primary authors for the Protocol as they will be best placed to share the methodology details, but the Corresponding Author of the present research paper should be included as one of the authors. By uploading your Protocols onto protocols.io, you are enabling researchers to more readily reproduce or adapt the methodology you use, as well as increasing the visibility of your protocols and papers. You can also

establish a dedicated workspace to collect your lab Protocols. Further information can be found at <https://www.protocols.io/help/publish-articles>.

With kind regards,

Daryl

Daryl Jason Verzosa David, PhD

Senior Editor, Nature Cell Biology
Advisory Editor, npj Biological Physics and Mechanics
Nature Portfolio

Heidelberger Platz 3, 14197 Berlin, Germany
Email: daryl.david@nature.com
ORCID: <https://orcid.org/0000-0002-9253-4805>

** Visit the Springer Nature Editorial and Publishing website at www.springernature.com/editorial-and-publishing-jobs for more information about our career opportunities. If you have any questions please click here.**